# ACL and HAT1 form a nuclear module to acetylate histone H4K5 and promote cell proliferation

Qiutao Xu [1,3], Yaping Yue [1,3], Biao Liu [1], Zhengting Chen [1], Xuan Ma [1], Jing Wang [1], Yu Zhao [1] & Dao-Xiu Zhou [1,2] ✉

Acetyl-CoA utilized by histone acetyltransferases (HAT) for chromatin modification is mainly generated by ATP-citrate lyase (ACL) from glucose sources. How ACL locally establishes acetyl-CoA production for histone acetylation remains unclear. Here we show that ACL subunit A2 (ACLA2) is present in nuclear condensates, is required for nuclear acetyl-CoA accumulation and acetylation of specific histone lysine residues, and interacts with Histone AcetylTransferase1 (HAT1) in rice. The rice HAT1 acetylates histone H4K5 and H4K16 and its activity on H4K5 requires ACLA2. Mutations of rice ACLA2 and HAT1 (*HAG704*) genes impair cell division in developing endosperm, result in decreases of H4K5 acetylation at largely the same genomic regions, affect the expression of similar sets of genes, and lead to cell cycle S phase stagnation in the endosperm dividing nuclei. These results indicate that the HAT1-ACLA2 module selectively promotes histone lysine acetylation in specific genomic regions and unravel a mechanism of local acetyl-CoA production which couples energy metabolism with cell division.

Histone acetylation is assumed to contribute to open chromatin by introducing structural barriers to compaction and by providing a platform for binding of specific reader proteins, thus playing critical roles in gene activation, DNA replication and repair, nucleosome assembly and dynamics, and cell cycle progression[1–4]. Histone acetylation is deposited by histone acetyltransferases (HATs) that transfer an acetyl group from acetyl coenzyme A (acetyl-CoA) to the lysine ε-amino groups on the N-terminal end of histones. Histone acetylation levels are associated with cellular growth and metabolic or energy states[5,6]. Fluctuation of cellular levels of acetyl-CoA, which is a key metabolic intermediate, affects histone acetylation in yeast and mammalian cells[7–10]. Acetyl-CoA used for histone acetylation is shown to be mainly generated by ATP citrate lyase (ACL) from citrate produced in mitochondrial TCA and exported to the cytosol[8,10,11], or locally produced in TCA in the nucleus[12]. Glucose is the primary substrate for acetyl-CoA generation by ACL in proliferating cells and ACL-sourced

acetyl-CoA contributes to increased histone acetylation during cellular response to high energy metabolism[8]. Recent results showed that ACL regulates the net amount of available acetyl groups, leading to alterations in acetylation of histone H3 lysine 9, 14 and 27 (H3K9/14/27) at the *MYOD* locus in myoblasts, thus increasing *MYOD* expression[13]. In plants, nucleocytoplasmic acetyl-CoA is also suggested to be mainly produced by ACL[14]. It was shown that increased levels of acetyl-CoA in mutants of the Arabidopsis acetyl-CoA carboxylase (that consumes acetyl-CoA) gene (*acc1*) led to increases of histone H3K27 acetylation, which is dependent on ACL-sourced acetyl-CoA[15]. However, how acetyl-CoA fluctuation affects acetylation of specific lysine residues in specific genomic domains remains unclear. A direct link between ACL and HATs in regulating histone acetylation has not been established yet.

The HAT family includes the General Control Non-repressible 5 (GCN5) -related N-terminal acetyltransferases (GNAT), cAMP-response-element-binding protein (CBP)/p300, the TATA-binding

[1]National Key Laboratory of Crop Genetic Improvement, Hubei Hongshan Laboratory, Huazhong Agricultural University, 430070 Wuhan, China. [2]Institute of Plant Science Paris-Saclay (IPS2), CNRS, INRA, University Paris-Saclay, 91405 Orsay, France. [3]These authors contributed equally: Qiutao Xu, Yaping Yue. ✉e-mail: dao-xiu.zhou@universite-paris-saclay.fr

protein-associated factor (TAF), and the MYST (MOZ, Ybf2/Sas3, Sas2, and Tip60) subfamilies. Histone AcetylTransferase1 (HAT1) belonging to the GNAT group is the founding member of the HAT family[16]. Originally, HAT1 was identified as a cytosolic enzyme (known as HAT-B) that acetylates newly synthesized histone H4 molecules at lysine residues 5 and 12 (H4K5 and H4K12) before their deposition in replicating chromatin and was therefore suspected of being involved in replication-dependent chromatin assembly[17]. Recent results showed that HAT1 is predominantly localized in the nucleus in mammalian cells[18]. HAT1 has multiple functions in chromatin processes but appears to be not required for nuclear import or deposition of core histones[18–20]. It is suggested that HAT1 senses cellular energy status or acetyl-CoA availability by placing available acetyl groups on nascent H4, which can be regenerated by deacetylation after importation in the nucleus to support nucleosomal histone acetylation[18]. Whether HAT1-sensing of cellular energy metabolism or acetyl-coa availability involves a direct recruitment of ACL remains unclear.

In this work, we found that ACL subunit A2 (ACLA2) is involved in histone acetylation at specific lysine residues by physically interacting with distinct HAT proteins. We showed that ACLA2 and HAT1 form a nuclear module to promote H4K5 acetylation at specific genomic regions, which is required for DNA replication, gene expression, and cell division in rapidly dividing sink (i.e. with high levels of sugar/glucose import) organs/tissues such as developing endosperm and root meristem. These results uncover a mechanism by which local acetyl-CoA production is established and maintained at specific genomic regions to stimulate high level histone acetylation required for cell proliferation, which integrates cellular nutrition/energy status to cell division decision.

## Results

### Loss of ACLA2 function affects cell division in developing endosperm and root meristem

Unlike animal ACL that has a monomeric structure plant ACL is split into two subunits, ACLA and ACLB[21,22]. ACLA contains the ATP-grasp (ATP-binding site) and citrate-binding domains, whereas ACLB harbors the CoA binding, CoA ligase, and citrate synthase domains (Supplementary Fig. 1a). Computation modeling indicated that ACLA and ACLB together recapitulated the human monomeric ACL 3D structure (Supplementary Fig. 1b). In rice, ACLA is encoded by three genes and ACLB by one gene (Supplementary Fig. 1a). Transient expression of ACLA2-GFP fusion protein in transfected tobacco leaf cells and immunostaining with an anti-ACLA2 polyclonal antibody of wild type rice root cells revealed that ACLA2 was localized in both the cytoplasm and nucleus (Supplementary Fig. 1c, d). Similarly, ACLB-GFP fusion proteins (under either 35S or the ACLB promoter) were observed in both the cytoplasm and nucleus of transfected tobacco leaf cells or rice leaf protoplasts (Supplementary Fig. 1c, d).

To study whether ACL activity affects histone acetylation in rice, we knocked out (KO) the three rice ACLA genes by CRISPR/Cas9. ACLA1 and ACLA3 KO plants were completely infertile (Supplementary Fig. 2a, b), as previously reported for acla1 mutants[23]. ACLA2 KO lines (acla2-1, 2, Supplementary Fig. 2c) were fertile (Supplementary Fig. 2c, d), but showed a much reduced seed setting rate and produced deformed and shrunken seeds (Fig. 1a, b). Histological analysis of developing seeds indicated that the development of embryo and endosperm was delayed in the mutants (Fig. 1c). During the early stage of endosperm development, cell proliferation takes place in a syncytium without cytokinesis, producing a liquid endosperm with free nuclei[24]. After a critical number of nuclei is produced, cellularization occurs and the tissue develops through regular cell divisions. The timing of cellularization is critical for nutrient storage and embryo provisioning, which can profoundly impact seed size and viability[25,26]. We found that the number of syncytial free nuclei per endosperm (at 1.5 day after fertilization, DAF) was decreased in the acla2 mutants

(Fig. 1d). The starch filling was reduced in the mutant endosperms at later stages of development (Fig. 1c, right).

At vegetative stage, acla2 plants showed reduced root lengths (Fig. 1e). The root meristem zone of the mutants had fewer cells and a shorter longitudinal length than the wild type (Fig. 1f). Staining with 5-ethynyl-20-deoxyuridine (EdU), which marks newly duplicated DNA, revealed a clear decrease of the replicating cell numbers in the acla2 roots compared to wild type, suggesting that the acla2 mutation impaired cell division in the root meristem (Fig. 1g). Besides, as observed previously[22], acla2 leaves produced brown lesion mimic patches (Supplementary Fig. 2d, right). Together, the observations suggested that the acla2 mutations impaired cell division in developing endosperm at the syncytium stage and in root meristem.

### The mutation of ACLA2 decreases nuclear acetyl-CoA levels and reduces acetylation of histone H3K14 and H4K5

Given the role of ACL in acetyl-CoA biosynthesis, we examined whether the acla2 mutation affected acetyl-CoA levels by using a recently reported method[27]. The acetyl-CoA levels in whole cell extracts of acla2 plant tissues were not significantly different from the wild type (Fig. 2a, left) suggesting that acetyl-CoA from non-ACL sources may be predominant in the whole cell extracts. However, in nuclear extracts of the acla2 mutants, the acetyl-CoA levels were significantly decreased compared to the wild type (Fig. 2a, right). We also tested the levels of citrate and glucose and detected similar levels of the metabolites in cytosol and nucleus. However, in acla2 plants, we detected higher citrate levels in both the cytoplasm and nucleus and higher glucose levels in the cytoplasm than wild type plants (Supplementary Fig. 3a), suggesting the mutation might alter accumulation of the metabolites. To test whether the decreased nuclear acetyl-CoA levels affected histone acetylation, we analyzed histone acetylation in developing seeds by immunoblotting and detected about 40–50% decreases of acetylation levels at H3K14 and H4K5 and 13–24% at H3K27 and H3K9 in the acla2 lines compared to wild type plants. Less or no change was detected at the other tested histone H3 or H4 lysine residues (Fig. 2b). The mutations had similar effects on H3K14, H3K27, and H4K5 acetylation in seedling roots (Fig. 2c), which were observed only in the nuclear but not cytoplasmic fraction (Supplementary Fig. 3b), in line with the reduced acetyl-CoA content in the nucleus. The results suggested that H3K14, H4K5 and, in a lesser extent, H3K27/9 acetylation was more sensitive to the loss of ACL activity than the other tested histone lysine residues.

### Interaction between rice ACLA2 and histone acetyltransferases

To study whether ACLA2 was involved in histone acetylation by selectively regulating specific HAT activity, we first tested protein interaction between ACLA2 or ACLB and the three rice GNAT family proteins[28] by yeast two-hybrid (Y2H) assays (Fig. 3a, b). The assays revealed that ACLA2 could interact with HAG703 and HAG704 (Fig. 3b). ACLA2 did not interact directly with HAG702 (homologous to GCN5), but instead with ADA2 (Fig. 3b). ADA2 forms a functional HAT complex with GCN5 which was shown to be required for H3K14 acetylation in rice and Arabidopsis[3,29]. HAG703 is homologous to ELP3, the function of which has been barely studied in plants. HAG704 is homologous to the yeast and human HAT1 (Fig. 3a). Interestingly, the ACLB subunit also interacted with HAG704 (Fig. 3b). Next, we tested ACL interaction with other HAT family members and found that ACLA2 could also interact with CBP/P300 and TAF family members (Supplementary Fig. 4), suggesting that ACLA2 may be recruited by several different HATs in rice cells. Unlike the yeast HAT1 that is localized in both cytoplasm and nucleus, HAG704 was found to be only localized in the nucleus of transient transfected tobacco cells when fused with GFP at the C-terminus (Fig. 3c). Immunostaining of rice seed and root cells with anti-HAG704 polyclonal antibody also detected HAG704 in the

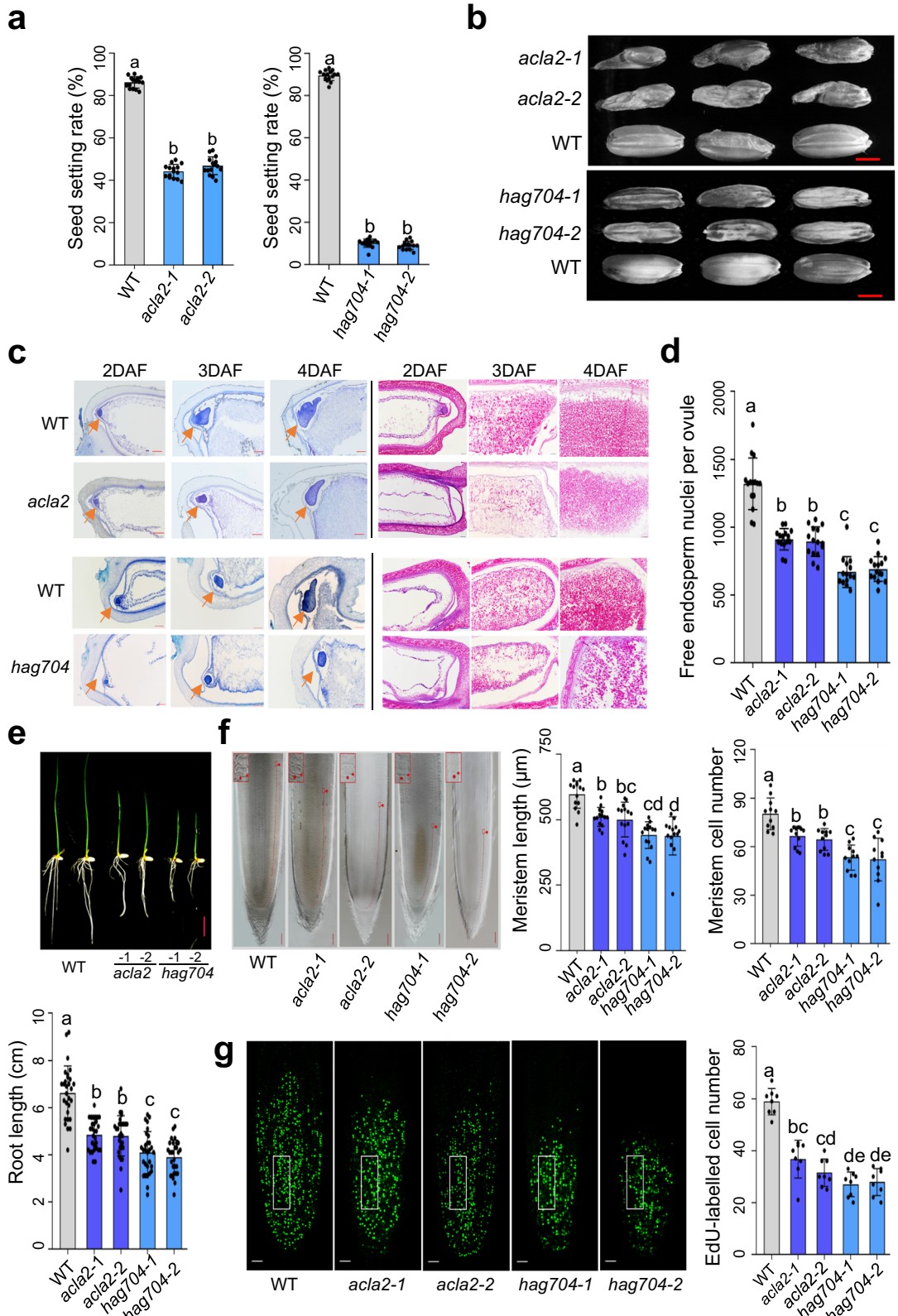

nucleus (Fig. 3c), suggesting that the rice HAT1 may function mainly in the nucleus.

The interaction between HAG704 and ACLA2 was confirmed by in vitro pull-down assays with *Escherichia coli*-produced HAG704-6×His-tag and ACLA2-GST proteins (Fig. 3d), bimolecular fluorescence complementation (BiFC) tests in transfected tobacco cells (Fig. 3e), and co-immunoprecipitation of the GFP-tagged ACLA2 and HAG704 proteins

expressed in rice protoplasts (Fig. 3f). The Y2H and BiFC assays with truncated segments revealed that the HAT domain of HAG704 and the citrate-binding domain of ACLA2 were required for the interaction (Supplementary Fig. 5a, b). Computer modeling confirmed the interaction between the subdomains (Supplementary Fig. 5c). The interaction was further confirmed by co-immunoprecipitation of cell extracts of rice developing seeds expressing HAG704-FLAG with

**Fig. 1 | Phenotypes of *acla2* and *hag704* mutants. a** Statistics of seed setting rates of wild type (WT), *acla2*, and *hag704* plants (*n* = 15). **b** The seed phenotype of *acla2*, *hag704*, and WT plants. Bars = 2 mm. **c** The paraffin section of WT, *acla2*, and *hag704* seeds. Left: embryo sizes (arrows) at 2–4 DAF. Right: endosperm development at 2–4 DAF. Starches are red stained. Bars = 100 µm. **d** Quantification of the syncytial endosperm nuclei (*n* = 15) in *acla2*, *hag704*, and WT at 1.5 DAF. **e** Root phenotype of WT, *acla2*, and *hag704* 7-d-old seedlings. Statistical differences of root lengths (*n* = 30) among different genotypes are shown in the lower panel. **f** Median longitudinal sections of WT, *acla2*, and *hag704* root tips. Arrows indicate the end of the meristem (from the quiescent center to the transition zone). Bars =

50 µm. Statistical analysis for meristem lengths (*n* = 14) and cell numbers (*n* = 11) in wild type, *hag704*, and *acla2* is shown on the right. **g** EdU-labeled cells in 4-d-old seedling root meristems of *acla2*, *hag704*, and wild type plants. Bars = 50 µm. Numbers of EdU-labeled cells in an arbitrary area of the root tip (white box, *n* = 8) were counted. EdU, 5-ethynyl-2′-deoxyuridine. For all statistics, the means ± SD of three independent biological replicates are shown. The significance was calculated using one-way ANOVA with Tukey's multiple comparison tests. Different letters on top of the bars indicate a significant difference (*p* < 0.05). DAF stands for day after fertilization. Source data are provided as a Source Data file.

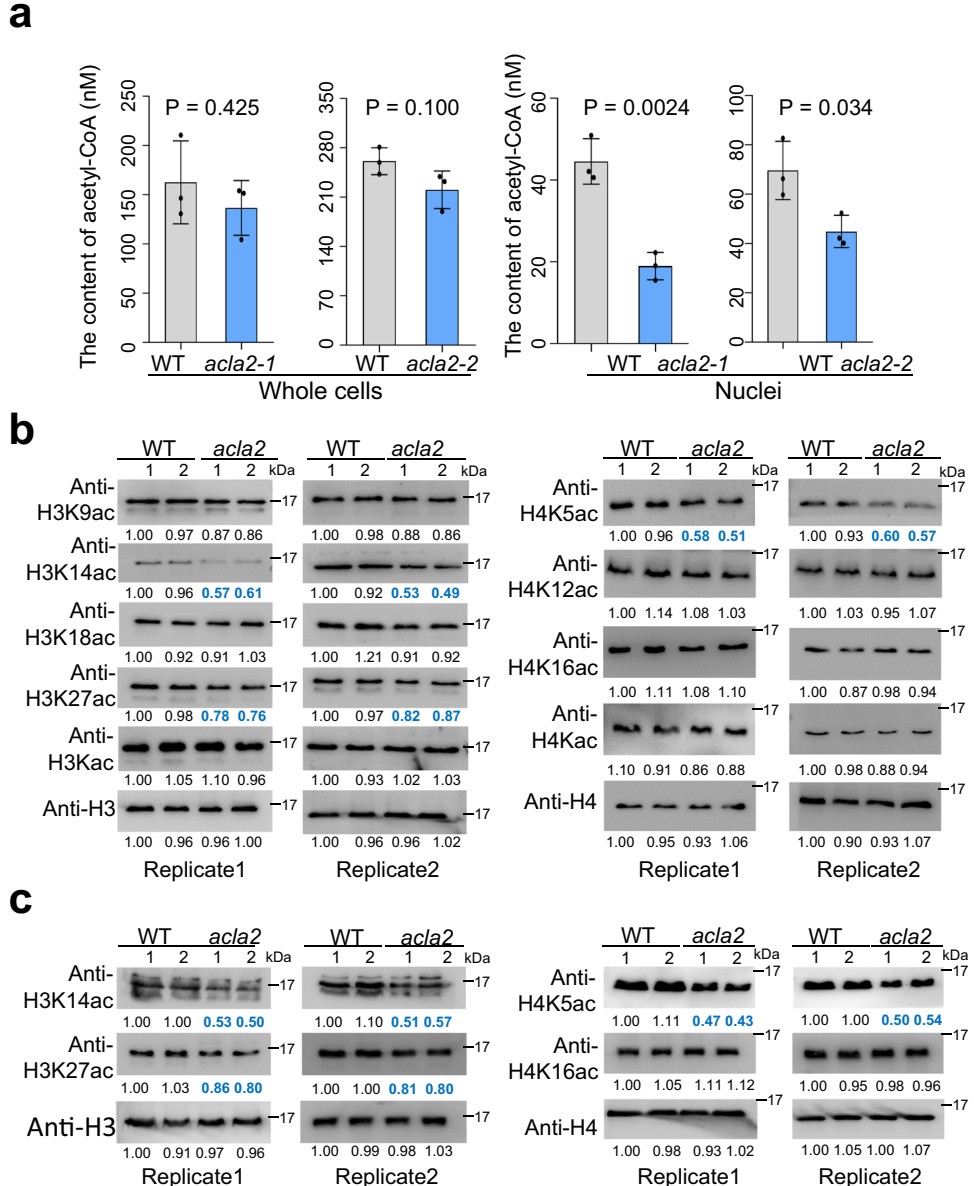

**Fig. 2 | Effects of the *acla2* mutations on cellular acetyl-CoA levels and histone acetylation. a** Acetyl-CoA level assays in wild type (WT) and *acla2* (*acla2-1* and *acla2-2*) mutant seedlings (7 d after germination, DAG). Significant difference was calculated by the two-tailed, paired Student *t* test. Error bars represent means ± SD (*n* = 3) from three biological replicates. **b** Analysis of histone acetylation levels of *acla2* (*acla2-1* and *acla2-2*) and wild type (WT-1 and WT-2) seeds by immunoblotting using anti-histone acetylation antibodies. Seeds (at 6 d after fertilization, DAF) were used to extract histone proteins. Total histone extracts were used for the analysis.

The H3 and H4 acetylation antibodies used for the tests are indicated on the left. Two replicates are shown. **c** Analysis of histone acetylation levels in wild type (two batches, WT1 and WT2) and *acla2-1* and *acla2-2* mutant seedlings (7 DAG) roots by immunoblotting using anti-histone acetylation antibodies. Total histone extracts were used for the analysis. Two replicates are shown. Immunoblotting bands were quantified using ImageJ and the relative signals indicated below each band were normalized with WT1 set as 1. Source data are provided as a Source Data file.

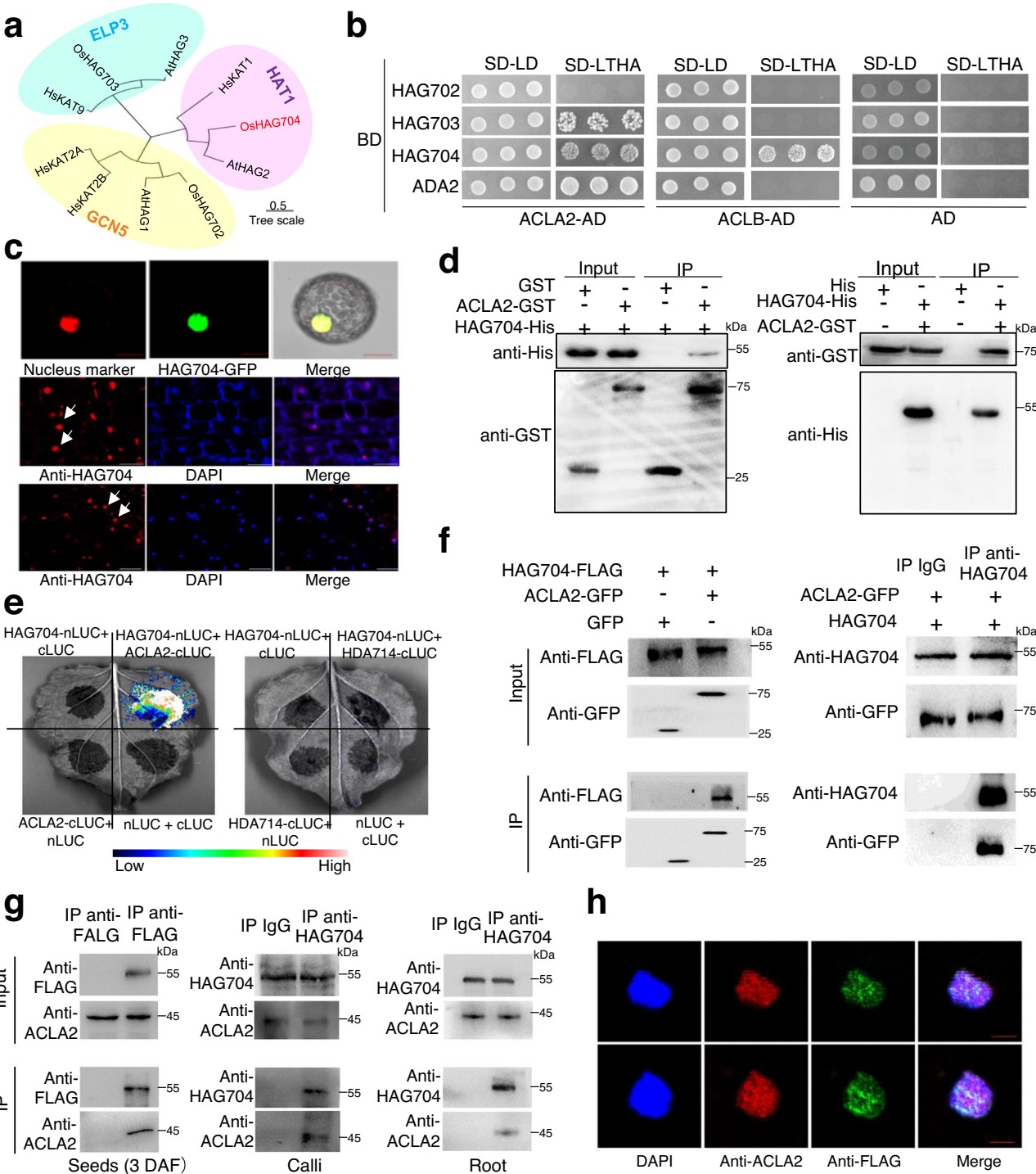

anti-FLAG antibody, or rice callus/root extracts by anti-HAG704, followed by immunoblotting with anti-ACLA2 and anti-HAG704 or anti-FLAG (Fig. 3g). Immunostaining of HAG704-FLAG expressing transgenic rice root nuclei with anti-ACLA2 and anti-FLAG antibodies indicated that both ACLA2 and HAG704 proteins were present in discrete and partially overlapped nuclear particles (Fig. 3h), suggesting that the proteins may interact and/or function in nuclear condensates.

### HAG704 loss-of-function results in seed and root developmental defects

To study the function of HAG704 and its relationship with ACLA2 in rice development, we produced *HAG704* CRISPR/Cas9 KO plants. Two KO lines (*hag704-1* and *hag704-2*) that showed no HAG704 protein

accumulation were selected for further analysis (Supplementary Fig. 2e, f). The KO plants showed a semi-dwarf phenotype at maturity (Supplementary Fig. 2f, left) and had a severely reduced seed setting rate and produced malformed and shrunken seeds (Fig. 1a, b). Although some *acla2* growth phenotypes (e.g. leaf lesion mimic) were absent, the *hag704* plants exhibited more severe seed developmental phenotypes than the *acla2* mutants, including retarded embryo growth, reduced number of the syncytial endosperm free nuclei (Fig. 1c, d), and decreased starch filling in the developing endosperm (Fig. 1c). In addition, the *hag704* mutants showed reduced root length, smaller root meristem sizes and cell numbers, and lower root cell proliferation rates, which were also more severe than the *acla2* mutants (Fig. 1e-g).

**Fig. 3 | HAG704 interacts with ACLA2 in vitro and in vivo. a** Phylogeny tree of GNAT proteins from *Arabidopsis* (At), *Oryza sativa* (Os), and *Homo sapiens* (Hs) generated by MEGA5 using the neighbor joining method. GNAT, general control non-repressible 5-related N-terminal acetyltransferase. ELP3, elongator acetyltransferase complex Subunit 3. HAT1, histone acetyltransferase 1. The scale bar indicates the average number of amino acid substitutions per site. **b** Yeast two-hybrid assays of interaction between GNAT family members and ACLA2 or ACLB. SD-LTHA is the selective medium for interaction. **c** Subcellular localization analysis of HAG704 protein. HAG704-GFP was localized in the nucleus in transfected tobacco protoplast (upper panels). Immunostaining with HAG704 antibody detected the protein also in the nucleus (arrows) of rice root (middle) and seed (lower panels) cells. Nuclei (blue) were stained with DAPI. The scale bars in upper, middle, and lower panels were 10 μm, 50 μm, and 50 μm, respectively. **d** Pull-down assays of HAG704 and ACLA2. Left: HAG704−6×His was incubated with GST or ACLA2-GST in GST beads and was pulled down from the ACLA2-GST conjugated GST beads. Right: ACLA2-GST was incubated with 6×His or HAG704-6×His in His beads and was pulled down from the HAG704-6×His conjugated His beads. **e** BiFC visualization of the interaction between HAG704-nLUC and ACLA2-cLUC proteins in the *N.benthamiana* epidermal cells. Histone deacetylase 714 (HDA714) was used as a negative control. **f** Co-immunoprecipitation assays of HAG704 and ACLA2 interaction in rice protoplasts. Left: 35S-HAG704-FLAG construct was cotransfected into rice protoplasts with 35S-ACLA2-GFP or GFP alone. Right: proteins extracted from rice protoplasts transfected with 35S-ACLA2-GFP and 35S-HAG704 were immuno-precipitated (IP) with anti HAG704 or IgG, and analyzed by immunoblotting with anti-HAG704 and anti-GFP. **g** Co-immunoprecipitation assay of HAG704 and ACLA2 interaction in rice tissues/organs. Left: Proteins isolated from developing seeds (3 DAF) of wild type and HAG704-FLAG expression plants were precipitated with anti-FLAG (left) and analyzed by immunoblots with anti-ACLA2. Middle: Proteins extracted from wild type rice calli were precipitated with anti-HAG704 or IgG as control and analyzed by immunoblots with anti-HAG704 and anti-ACLA2. Right: Proteins extracted from wild type root tip were precipitated with anti-HAG704 or IgG as control and analyzed by immunoblots with anti-HAG704 and anti-ACLA2. **h** Co-immunofluorescence of HAG704 and ACLA2 in rice root nuclei. HAG704-FLAG expressing transgenic rice root nuclei were immunostained with both anti-FLAG (green) and anti-ACAL2 (red) antibodies. Nuclei (blue) were stained with DAPI. Bars = 5 μm. For Fig. 3d, f, and g, the experiments were repeated at least two times with similar results, and representative data are presented. For Fig. 3c and h, the experiments were repeated three times with similar results, and representative data are presented. Source data are provided as a Source Data file.

To study the genetic relationship, we produced the *hag704/acla2* double mutants by genetic crosses of *acla2-1* and *hag704-1* plants. The phenotypes of the double mutants were more severe than *hag704-1* plants (Supplementary Fig. 2g). The observations suggested that in addition to HAG704, ACLA2 was involved in additional pathways, consistent with the interaction of ACLA2 with several different HAT family members and with the *acla2* mutation effects on acetylation of different lysine residues.

## HAG704 is required for histone H4K5 and H4K16 acetylation in rice cells

Immunoblotting analysis of developing seeds revealed clear decreases of acetylation levels at H4K5 and H4K16 in the *hag704* plants (Fig. 4a). There was no clear change for acetylation at H4K12 and the tested H3 lysine residues. The results suggested that HAG704 targeted specifically histone H4 but at different lysine sites from yeast or human HAT1. The decreases of H4K5 and H4K16 acetylation in seeds were also detected in *hag704* root cells (Supplementary Fig. 6a), which were, however, detected only in nuclear but not cytoplasmic histones (Supplementary Fig. 6b). The data suggested that unlike the yeast or mammalian HAT1 that acetylates newly synthesized histone H4 K5 and H4K12 in the cytoplasm, HAG704 rather acetylates histone H4K5 and H4K16 in the nucleus, consistent with its nuclear localization (Fig. 3c).

To confirm the results, we tested HAG704 histone acetyltransferase activity by co-transfecting tobacco leaf cells with the vectors expressing 6×His-tagged histone H3 and H4 proteins and the vector expressing HAG704 or the control (empty) vector. The tagged histone H3 or H4 proteins were isolated by His-tag affinity and tested for acetylation by immunoblotting (Fig. 4b). The co-transfection with HAG704 resulted in acetylation of histone H4K5 and H4K16, but not H4K12, in tobacco cells (Fig. 4b), confirming the *hag704* mutation effects in rice seed and root cells. The tested H3 lysine residues showed acetylation regardless of the presence or absence of HAG704 (Fig. 4b), which was likely due to endogenous HAT activities in the transfected cells. The results confirmed that HAG704 had a HAT activity at H4K5 and H4K16.

## ACLA2 stimulates HAG704 HAT activity at histone H4K5

Since both the *hag704* and *acla2* mutations reduced H4K5 acetylation, we tried to determine whether ACLA2 was involved in the HAG704 HAT activity. We produced His-tagged histone H4 together with or without combinations of the vectors expressing HAG704/ACLA2/ACLB-FLAG fusions in tobacco leaf cells. His-targeted histone H4 protein was purified and its acetylation level was analyzed by immunoblots using antibodies against acetylated histone H4K5 and H4K16. The results showed that the presence of HAG704 increased H4K16 acetylation by >5 folds, but only 1.5−2 folds for H4K5 acetylation (Fig. 4c). However, the presence of the ACL holoenzyme (ACLA2 and ACLB), but not with ACLA2 or ACLB subunit alone, further elevated the H4K5 acetylation level by 3–4 times (Fig. 4c). The effect of ACLA2 on HAG704-mediated H4K16 acetylation was less important (<30% increase) (Fig. 4c). The data corroborated the effects of *acla2* and *hag704* mutations on H4 acetylation, and indicated that ACLA2 selectively stimulated the HAG704 HAT activity at H4K5. The similar effect of the double and single mutations on H4K5 acetylation (Supplementary Fig. 6c) supported the hypothesis. Computer modeling of the HAG704-ACL-H4 complex structure suggested a close contact of the ACLB subunit with H4K5 (Supplementary Fig. 5d).

## The *acla2* and *hag704* mutations have similar effects on transcriptome and genome-wide H4K5 acetylation in dividing nuclei of the syncytial endosperm

To further study the functional relationship of ACLA2 and HAG704 in genome-wide histone acetylation and gene expression in a specific cell type, we isolated the syncytial endosperm dividing nuclei at 1.5 DAF by capillary aspiration from the wild type and the mutant plants (see Methods). First, RNA-sequencing (RNA-seq) was performed with about 30,000 syncytial endosperm nuclei collected per sample to examine differentially expressed genes (DEGs) in *hag704* and *acla2* relative to wild type (Supplementary Fig. 7a, b). Three biological replicates were tested for each sample (Supplementary Fig. 7c, Supplementary Data 1). Using the TissueEnrich software[30], from the wild type nuclei we identified 1260 genes specially expressed in the endosperm (Supplementary Fig. 8a, left, Supplementary Data 1), which were enriched for translation, nitrogen compound metabolic process, and gene expression pathways (Supplementary Fig. 8b). Among these genes, 174 were specifically expressed at the syncytium stage (Supplementary Fig. 8a, right, Supplementary Data 1). These genes were enriched for phosphorelay signal transduction system, cytokinin-activated signaling network, and response to stimulus pathways (Supplementary Fig. 8c), consistent with previous results showing correlation between cytokinin levels and endosperm cell division activity[31].

Compared to the wild type, the *hag704* and *acla2* mutations resulted in 1248 and 1585 down-regulated genes and 762 and 1083 up-regulated genes (>2-fold, P < 0.05) in the syncytial endosperm nuclei, respectively (Supplementary Fig. 9a, Supplementary Data 2). The DEGs in the *hag704* and *acla2* mutants were highly overlapped (hypergeometric tests, P < 1.203e−160 and P < 1.239e−160), as 72% (902/1248) of the down-regulated and 64% (488/762) of the up-regulated genes in *hag704* were found to be also down or up-regulated in the *acla2*

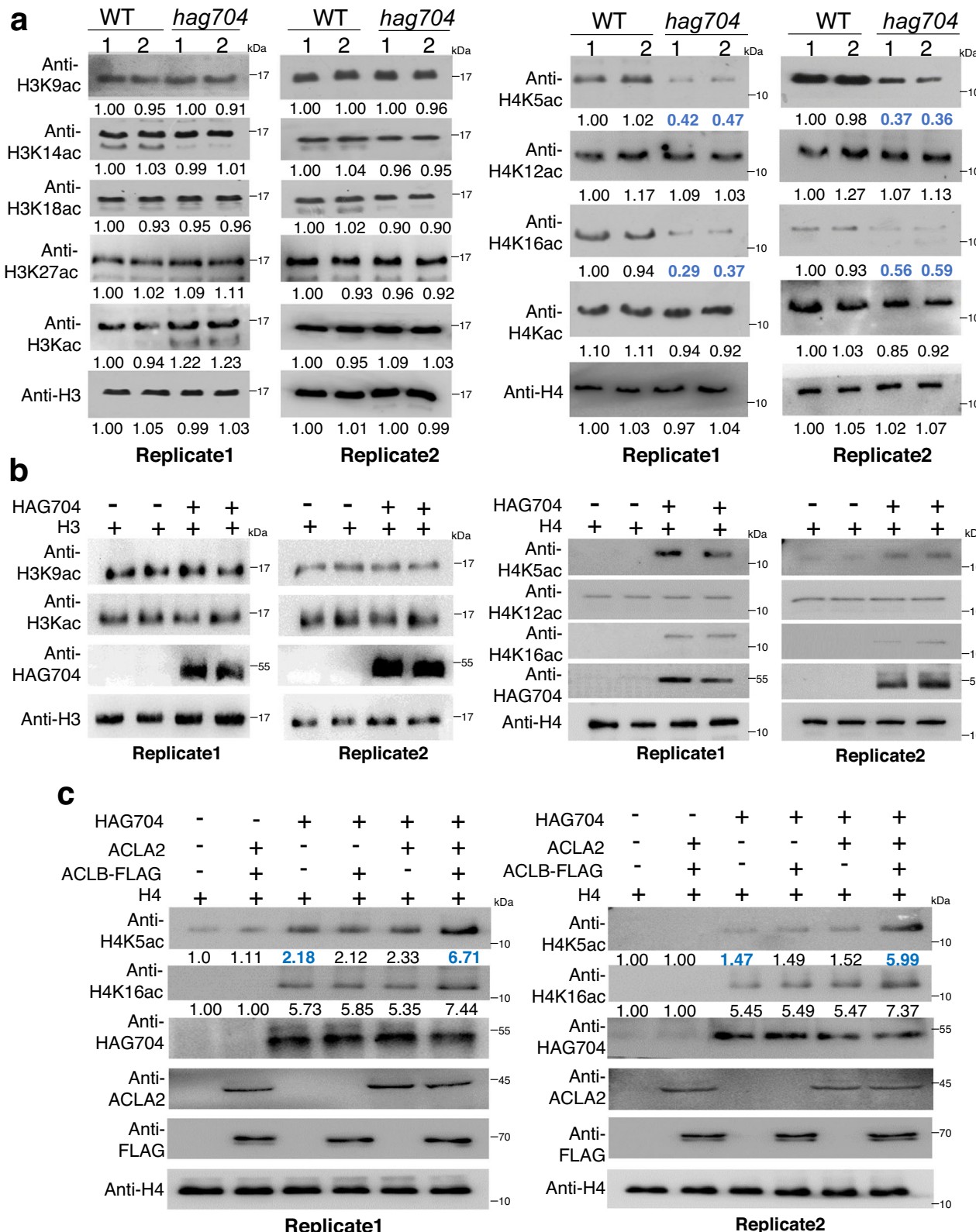

mutants (Supplementary Fig. 9b), indicating that HAG704 and ACLA2 regulated largely a same set of genes in the syncytial endosperm nuclei. The 902 downregulated genes in both mutants were enriched for cellular metabolic, oxidation-reduction, transcription regulation, and starch metabolic processes (Supplementary Fig. 9c).

Next, we employed Cleavage Under Targets and Tagmentation (CUT&Tag), a technology for epigenomic profiling of small samples

and single cells[32], to compare genome-wide acetylation of H4K5 and H4K16 in the wild type, *hag704*, and *acla2* syncytial endosperm nuclei. For each CUT&Tag sample, about 30,000 endosperm free nuclei were collected and two biological replicates were carried out (Supplementary Fig. 7d, Supplementary Data 3). In the wild type, 26, 602 peaks (18,997 marked genes) of H4K5 acetylation and 23, 937 peaks (16,022 marked genes) of H4K16 acetylation were identified. Both histone

**Fig. 4 | HAG704 HAT activity on H4K5 depends on ACL. a** Analysis of histone acetylation levels in *hag704-1* and *hag704-2* mutants compared with the wild type (WT-1 and WT-2) plants by immunoblotting. Histone proteins extracted from 6 DAF seeds were tested. The histone acetylation antibodies used in the tests are indicated on the left. Total histone extracts were used for the analysis. Two replicates are shown. The immunoblot signals were quantified using ImageJ. Relative signals (to the control WT1 signal set at 1) are indicated below the bands. **b** In vitro HAG704 histone acetyltransferase activity assays. His-tagged histone H3 or H4 proteins were produced alone or together with and HAG704 protein in co-transfected tobacco leaf cells. After purification with His resins, the acetylation levels of the His-tagged histone H3 or H4 proteins were analyzed by immunoblotting with the antibodies indicated on the left. Anti-H3 and anti-H4 antibodies were used as loading controls. **c** ACL stimulated the HAG704 histone acetyltransferase activity at H4K5. His-tagged H4 protein was produced together with or without combinations of HAG704, ACLA2 and ACLB-FLAG in tobacco leaf cells. The acetylation level of purified histone H4 was analyzed by immunoblots using the H4K5ac and H4K16ac antibodies. ACLA2, HAG704, and ACLB were detected by anti-ACLA2, anti-HAG704, and anti-FLAG antibodies respectively. Anti H4 antibody was used as loading controls. Two replicates are shown. Immunoblotting results were quantified using ImageJ. Relative signals (to controls, i.e., in the absence of HAG704, set at 1) are indicated below the bands. Source data are provided as a Source Data file.

marks were enriched in genes with H4K5 acetylation highly peaked at the transcription start sites (Supplementary Fig. 10a).

Consistent with the immunoblot results (Figs. 2b, 4a), the overall H4K5 acetylation level was lower in the *hag704* and *acla2* endosperm nuclei than the wild type (Fig. 5a), while the overall H4K16 acetylation level was decreased in *hag704* but unchanged in *acla2* (Fig. 5b). The *hag704* and *acla2* mutations resulted in a decrease of H4K5 acetylation in 4453 (3209) to 5480 (4485) peaks (genes) and an increase of the mark in about 600 peaks (>1.5 fold, *P* < 0.05) (Fig. 5c, Supplementary Data 3). The changes of H4K5 acetylation levels in both mutants (versus WT) showed a high correlation (*r* = 0.73, Fig. 5d). In fact, 58.61% peaks (64.16% genes) with H4K5 acetylation loss in *hag704* overlapped with those in *acla2* (Fig. 5e), indicating that HAG704 and ACLA2 largely targeted the same genomic loci for H4K5 acetylation in the syncytial endosperm nuclei. The *hag704* mutation led to a loss of H4K16 acetylation in 1825 (1512) peaks (genes) and a gain of the mark only in 136 peaks (Fig. 5f, left), while the *acla2* mutation resulted in a loss of H4K16 acetylation in 503 (284) peaks (genes) and a gain of the mark in 234 peaks (Fig. 5f, right, Supplementary Data 3), consistent with the unchanged overall H4K16 acetylation level in the *acla2* mutants (Fig. 2b). The differential H4K5 and H4K16 acetylation peaks in *hag704* showed a low level of correlation or overlap (Supplementary Fig. 10b, c), suggesting that HAG704 targeted the two histone H4 lysine residues for acetylation by different mechanisms. The data confirmed that HAG704 and ACLA2 were both required for nucleosomal H4K5 acetylation.

Scattering plot analysis indicated that about 58–60% (726/1,248 in *hag704*, 953/1,585 in *acla2*) of the down-regulated genes showed loss of H4K5 acetylation (>1 fold, *P* < 0.05) in the mutants relative to wild type (Fig. 5g). By contrast, changes of H4K16 acetylation did not show a clear overlap with gene expression changes in *hag704* (Supplementary Fig. 10d). Among the 902 genes downregulated in both *hag704* and *acla2*, 286 lost H4K5 acetylation (>1.5 fold, *P* < 0.05) in *hag704*, 162 lost the mark in *acla2*, and 136 in both mutants (Supplementary Fig. 10e, Supplementary Data 4). These genes were enriched in starch biosynthetic, glycogen biosynthetic, and carbohydrate metabolic processes (Supplementary Fig. 10f). The analysis suggested that HAG704 and ACLA2-dependent H4K5 acetylation preferentially targeted genes involved in starch or glucose polymer biosynthesis and energy metabolism in developing endosperm, which was consistent with the endosperm starch filling and cell division defects of the mutants (Fig. 1c). RT-qPCR and ChIP-qPCR tests of a selection of 9 genes confirmed their down-regulation of expression and H4K5 acetylation in the mutant endosperm (Fig. 6, Supplementary Fig. 11). The double mutations had no additive effect on the expression or acetylation levels (Fig. 6), suggesting that HAG704 and ACLA2 were both required for H4K5ac on these genes.

### HAG704 and ACLA2 mutations affect DNA synthesis during cell cycle
Immunostaining of syncytial endosperm nuclei with HAG704 and ACLA2 antibodies also detected the proteins in small nuclear particles (Fig. 7a). Aside from regulating gene transcription, histone H4 acetylation has also been reported to be involved in DNA replication and cell division[33]. To test whether the HAT1-ACLA2 module-regulated H4K5 acetylation affected DNA replication, using flow cytometry, we analyzed the DNA contents of the syncytial endosperm nuclei from wild type and *acla2*, *hag704* and the double mutant plants (Supplementary Fig. 12). As shown in Fig. 7b, the *hag704* and *aclal2* mutations caused a blockage of G2/M-phase progression with higher accumulation of nuclei at S phase than the wild type (Fig. 7b, c). The *hag704 acla2* double mutant showed similar effects to the single mutants (Fig. 7b, c). These data suggested that, in addition to gene expression, HAG704 and ACLA2-dependent histone H4K5 acetylation was required for DNA replication in rapidly dividing nuclei of the syncytial endosperm.

## Discussion
### ACLA2 interacts with different HATs to selectively regulate histone lysine acetylation sites
Previous work showed that cellular changes in the acetyl-CoA concentration, which reflect cellular nutrient status, affect the levels of histone acetylation[34,35]. It remains unclear how exactly acetyl-CoA pool is locally produced and how acetyl-CoA availability influences site-specific histone acetylation. The requirement of ACLA2 for nuclear acetyl-CoA production and acetylation of several histone lysine residues and the ACLA2 interaction with different HAT family members are in favor of the hypothesis that the recruitment of ACLA2 by different HAT complexes is a mechanism to establish locally high concentrations of acetyl-CoA required for acetylation at specific lysine sites of histones. In plants, the GCN5-ADA2 HAT complex is required for H3 acetylation, particularly at H3K14[29,36,37]. The *acla2* mutation effects on acetylation of histone H3 lysine sites and the ACL-ADA2 interaction imply that GCN5-ADA2 -dependent acetylation at H3K14 and, to a lesser extent, at H3K27/K9 requires the recruitment of ACLA2 for local acetyl-CoA production. This is consistent with previous results that suppression of both ACL and GCN5 in mammalian cells did not lead to additive inhibition of acetylation of the histones examined[8]. The observations that HAG704 (HAT1)-mediated acetylation at H4K5 but not H4K16 involves ACLA2 and that the *acla2* and *hag704* single and double mutations had similar effects on H4K5 acetylation allow assuming that HAT1-mediated H4K5 acetylation that is needed to support rapid cell division requires a high rate of acetyl-CoA supply. However, it is not excluded that the ACLA2 interaction may influence HAT enzymes targeting to specific histone lysine residues. The presence of both ACLA2 and HAG704 (HAT1) proteins in nuclear particles raises the possibility that their interaction may take place in nuclear condensates, which may locally increase the acetyl-CoA concentration in chromatin domains. Conversely, HAT1 and ACLA2-mediated histone acetylation may facilitate the formation of such condensates, as previous work showed that histone acetylation and bromodomain-containing "reader" proteins alter or re-induce in vitro droplets formed by nucleosomal arrays[38]. In addition, the histone lysine acetylation "reader" BRD4 seems to form condensates regulating cell-identity genes in mouse embryonic stem cells[39].

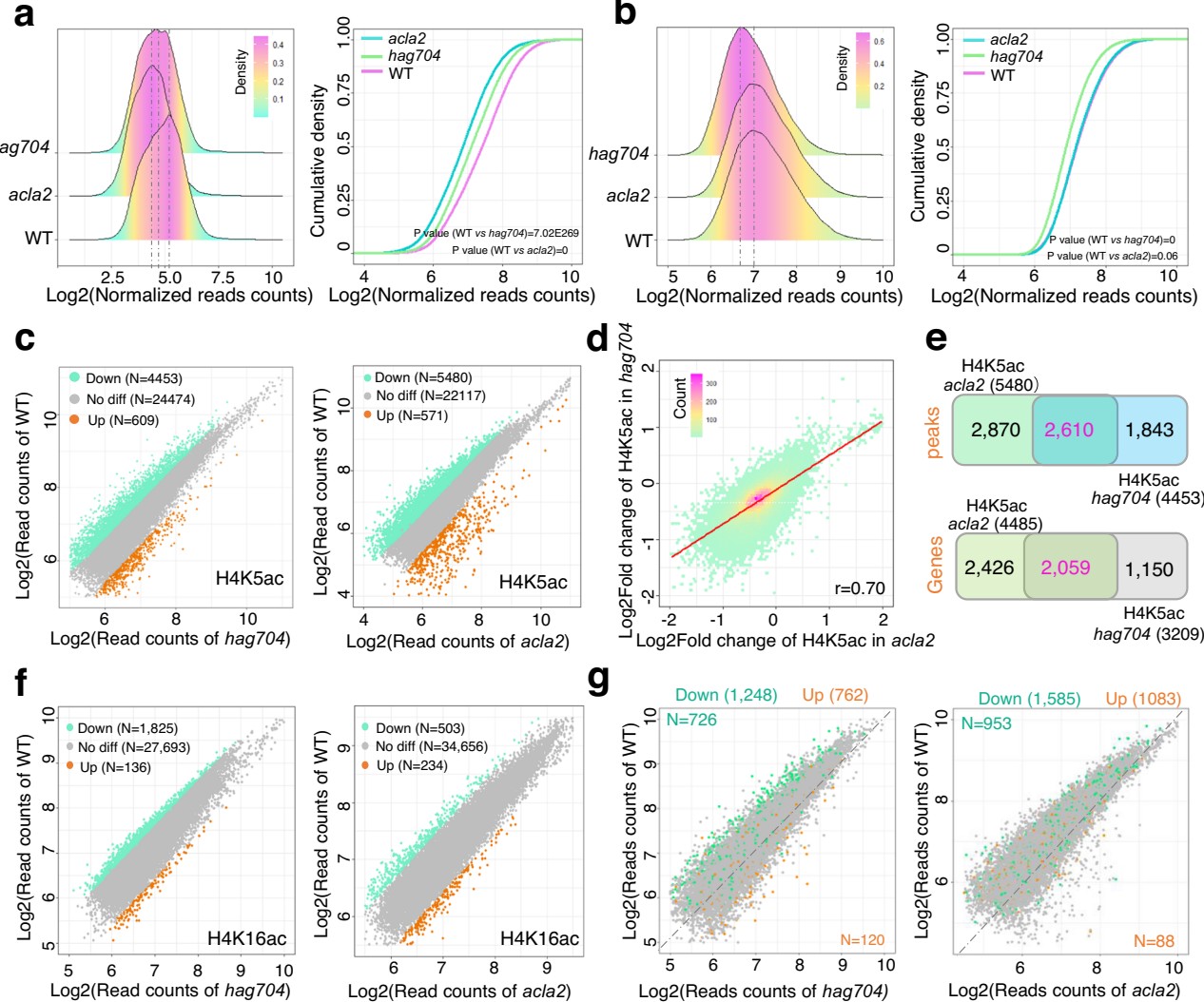

**Fig. 5 | Genome-wide analysis of H4K5ac and H4K16ac in *hag704* and *acla2* the syncytial endosperm nuclei.** Ridge plots (left) and cumulative density plots (right) of H4K5 (**a**) and H4K16 (**b**) acetylation levels in wild type, *hag704*, and *acla2* endosperm nuclei at the syncytium stage (1.5 DAF). *P* value was calculated by a two-tailed, paired Student *t* test. **c** Scatter plot of ChIP-seq data showing the differential occupancy of H4K5 acetylation in *hag704* (left) or *acla2* (right) endosperm nuclei relative to WT. Peaks with fold changes >1.5 and *p* value < 0.05 were considered as up or down-regulated. **d** Correlation between changes in H4K5 acetylation ChIP-seq signals in *hag704* and *acla2*. Correlation analysis was based on the differential H4K5ac peaks commonly detected in *acla2* vs WT and *hag704* vs WT (*N* = 26,995). Pearson coefficient was shown. **e** Venn diagrams showing the overlap of down-regulated H4K5 acetylation peaks (upper) or genes (lower) in *acla2* and *hag704*.

**f** Scatter plots of H4K16 acetylation ChIP-seq data in the endosperm nuclei between *hag704* (left) or *acla2* (right) and the WT. Peak numbers with fold change > 1.5 and p value < 0.05 are indicated by colors. **g** Localization of the *hag704* and *acla2* mutant DEGs in the scattering plots of H4K5 acetylation in the *hag704* and *acla2* mutants versus the wild type. Green and brown dots represent respectively downregulated and upregulated genes in the mutants. The numbers (N) of genes that showed down (green) or up (brown) regulation for both expression (fold change >2, *p* < 0.05) and H4K5 acetylation fold change (>1, p < 0.05) are indicated. For Fig. 5c, f, and g, the p-values were derived from a two-sided, unpaired Wald test without multiple comparison correction. Source data are provided as a Source Data file.

## Function of the HAT1-ACLA2 module-mediated H4K5 acetylation in cell division

The majority of acetyl groups destined for incorporation into chromatin modifications are produced de novo from glucose[8,40], the primary nutrient utilized by proliferating cells. Previous results showed that human HAT1 captures glucose-derived acetyl groups to acetylate nascent H4, rather than nucleosomal histones[18]. Nascent H4 is deacetylated upon assembly into chromatin[41,42], to regenerate nuclear acetyl-CoA that can be used by downstream HATs[18]. The present data showing that the HAG704 has a function to acetylate histone H4K5 and H4K16 and that, unlike yeast and mammalian homologs that acetylate H4K5 and H4K12 of newly produced histones in the cytoplasm, HAG704 is not involved in H4K12 acetylation indicate that the rice HAT1 has a divergent function. No clear effect of the *hag704* mutation

on acetylation of cytoplasmic histones supports this hypothesis. The HAG704 function to regulate acetylation of nucleosomal histones suggests that the plant HAT1 may directly integrate cellular energy metabolism to chromatin acetylation to regulate DNA replication and/or gene expression.

The similar effects of the *acla2* and *hag704* mutations on endosperm and root development suggest that the HAT1-ACLA2 module-mediated H4K5 acetylation integrates high glucose levels to cell division regulation in these rapidly growing sink organs to which sugar molecules are actively imported. Acetylation of histones is regulated in a cell-cycle dependent manner and can contribute to normal cell cycle progression. The S phase stagnation phenotype of the *acla2* and *hag704* endosperm nuclei points to a role of H4K5 acetylation in cell cycle progression, consistent with previous results that H4 acetylation

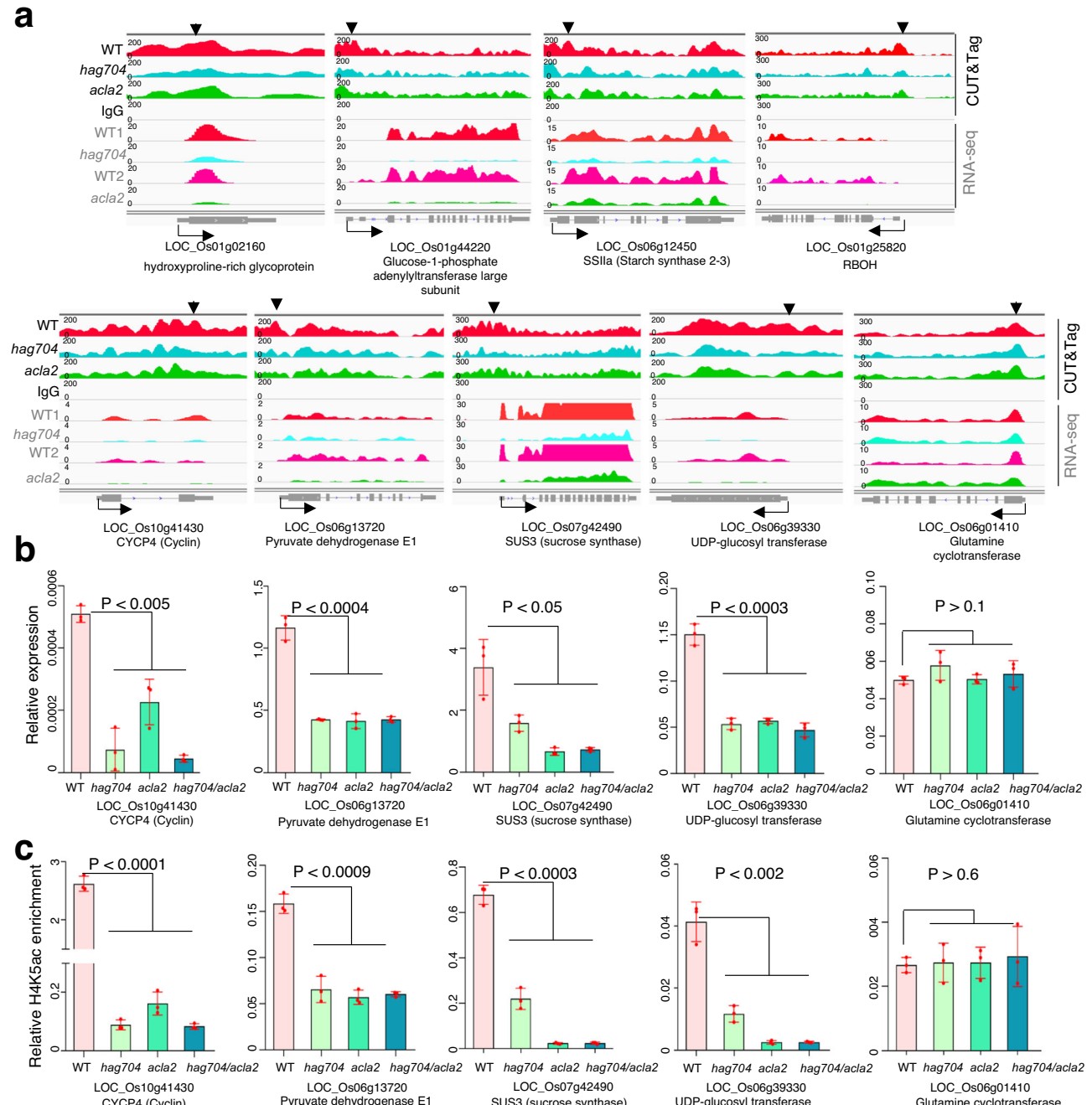

**Fig. 6 | Validation of expression and H4K5 acetylation of genes commonly regulated by HAG704 and ACLA2. a** Integrative Genomics Viewer (IGV) screenshots showing H4K5 acetylation peaks and transcript levels of 9 genes in WT, *hag704*, and *acla2* endosperm nuclei. **b** qRT-PCR analysis of transcript levels in 4 selected genes (relative to the *Actin* transcripts) in three DAF seeds of WT, *hag704*, *acla2*, and the *acla2 hag704* double mutants. **c** H4K5 acetylation ChIP-pCR assays of

3 DAF seed chromatin isolated from WT, *hag704*, *acla2*, and *acla2 hag704* plants. For all the data, bars indicate means ± SD from three replicates. *P* value was calculated by a two-tailed, paired Student *t* test. DAF, days after fertilization. Analysis of the other genes is shown in Supplementary Fig. 11. Source data are provided as a Source Data file.

is required for DNA replication[33]. The *acla2* and *hag704* mutation effects on cell division may also be related to decreased expression of cell proliferation-related genes that are dependent on H4K5 acetylation. This is supported by the observations that in the syncytial endosperm dividing nuclei the *acla2* and *hag704* mutant DEGs are not specific to the endosperm function or the developmental stage, but enriched for cell division, carbon metabolism and starch biosynthesis. Both the endosperm and the root meristem are non-photosynthetic sink organs to which photosynthetic products (sugar molecules) are actively imported to support rapid cell division at the early

developmental stages and storage in differentiated cells at later stages. The HAT1-ACLA2 module-mediated H4K5 acetylation may represent a cellular response to high energy metabolism in the developing endosperm and root meristem. The enrichment for carbohydrate metabolic function of the (expression and H4K5 acetylation) downregulated genes in both the *acla2* and the *hag704* mutants suggests that the nutrient-responsive H4K5 acetylation selectively impacts the expression of genes required to reprogram intracellular metabolism to utilize glucose for ATP production and macromolecular synthesis. Collectively, our data indicate that the HAT1-ACLA2 module-mediated H4K5

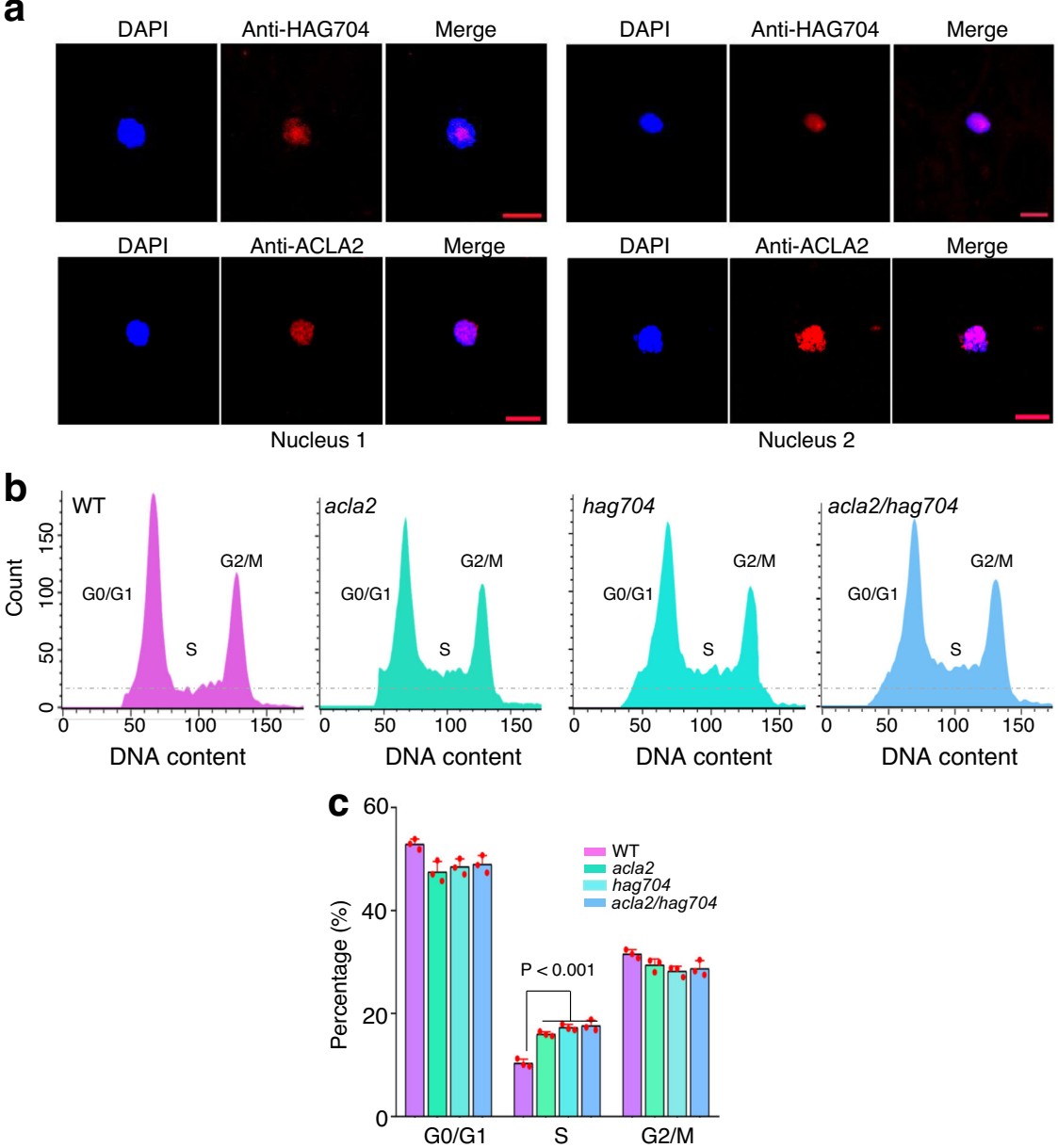

**Fig. 7 | The *hag704* and *acla2* mutations caused S-phase stagnation in the dividing nuclei of the syncytial endosperm. a** Immunofluorescent staining of endosperm nuclei with anti-HAG704 (upper) and anti-ACLA2 (lower) antibodies. Nuclei (blue) were stained with DAPI. Bars = 10 μm. The experiments were repeated three times with similar results, and representative data are presented. **b** Representative histograms depicted cell cycle distribution in the endosperm nuclei of WT, *acla2*, *hag704*, and *acla2 hag704* plants. The endosperm nuclei (at 1.5 DAF) were isolated by capillary, stained with DAPI, and analyzed by flow cytometry. **c** The percentages of the total nuclei population in each phase of the cell cycle are represented. Bar indicates mean ± SD from three replicates. *P* value was calculated by a two-tailed, paired Student *t* test. Source data are provided as a Source Data file.

acetylation coordinates cellular nutrient/energy metabolism/state to cell division decision by regulating DNA replication and gene expression.

## Methods

### Plant materials and growth conditions
In this study, the rice variety Zhonghua11 (*Oryza sativa* spp. *Japonica*) was used to produce the transgenic plants. The CRISPR/Cas9[43] mutants were produced by transformation of the binary T-DNA vectors containing Cas9 and sgRNA into rice calli. The mutation was decoded by DSDecode[44]. The over expression transgenic plants were produced using the maize (*Zea mays*) ubiquitin promoter vector pU1301-3×FLAG. The *hag704* and *acla2* double mutants were created through a genetic cross between *hag704-1* mutant and *acla2-1* mutant plants. For in vitro cultures, seeds were surface-sterilized and germinated in media containing 0.3% phytagel supplemented with 2% (w/v) sucrose at 28 °C (in light) and 24 °C (in dark) with a 14 h light/10 h dark cycle. Tobacco (*Nicotiana benthamiana*) plants used for transient expression were grown in soil for 6 weeks at 20 ± 2 °C, 8 h photoperiod, and 100 μmol quanta/ (m² s) illumination. All indicated rice plants were grown in the Wuhan area during the summer rice growing season.

### Histochemical analysis
Rice seeds were collected and fixed in FAA (4% formaldehyde, 10% acetic acid, and 50% ethanol) before being dehydrated in a series of graded concentrations of ethanol. These seeds were polymerized at

37 °C for three days before being embedded in paraffin. The embedded samples were first cut into 10 μM thick sections and counterstained with periodic acid-Schiff reagent and toluidine blue before being examined using a Nikon Ni-E light microscope with a CCD camera.

## ACLA2 and HAG704 Antibodies production

To produce the ACLA2 antibody, *E. coli*-produced full-length ACLA2 was used to immunize rabbits, and antiserum was collected after five doses of immunization. For the HAG704 antibody, rabbits were immunized with two synthesized peptides (KQKGTDAAADPKKRR and DADDQTVEQQPEDLKTQE) coupled to KLH, and antiserum was collected after five doses of immunization.

## Subcellular localization assays

To make the GFP fusions, the full length cDNAs of *HAG704* and *ACLA2* were subcloned into the pCambia1301 vector. The AgI-PrI method was used for transient expression and protoplast isolation[45]. Using a syringe without a needle, *A. tumefaciens* cells were infiltrated into the abaxial epidermis of 6-week-old tobacco (*N. benthamiana*) leaves. After 48 h of transfection, a laser confocal microscope (Olympus FV1200) was used to photograph leaf cells or isolated protoplasts.

For ACLB subcellular localization, full length cDNA of ACLB and its promoter (2 kb) were first subcloned into the pCambia1300 vector to make the GFP fusion. The fusion was transiently expressed in rice protoplasts using polyethylene glycol-mediated transformation and the subcellular localization was visualized using a confocal laser scanning microscope (Olympus FV1200) after overnight incubation.

## Edu staining

EdU staining was conducted using an EdU kit (Ribobio, C10310) as described previously[3]. Crown roots of three-day-old (post-germination) seedlings were submerged in a 50 μM EdU solution for 3 h. After 30 min of fixation in 4% paraformaldehyde, a longitudinal vibration slice was collected and treated with Apollo. The florescence was detected with a laser confocal microscope (Olympus, FV1200).

## Rice endosperm free nuclei isolation

The rice syncytial endosperm nuclei isolation was performed following a previous report[24]. In general, the ovary at 1.5 DAF was initially sliced with razor blades. The endosperm nuclei were then sucked out using a glass micropipette (1 mm in diameter) with a tip aperture of roughly 25–30 μm. Finally, the nuclei were transferred into 1.5 ml RNase-free tubes containing 200 μl PBS (pH 7.4) and stored for CUT&Tag assays or RNA extraction.

## Cell cycle analysis

Syncytial endosperm nuclei were isolated at 1.5 DAF, labeled with DAPI (4′, 6-diamidino-2-phenylindole dihydrochloride), and subjected to flow cytometry analysis using a FACS Calibur system (BD FACSCelest, USA) to investigate the cell cycle. At least 10,000 stained nuclei per sample were analyzed.

## Acetyl-CoA, glucose, and citrate contents assays

The concentration of acetyl-CoA was measured as previously described[27]. Seven-day-old rice tissues (0.1 g, green part) were crushed and suspended in 0.9 mL of PBS (pH 7.0). The suspensions were vortexed and then placed at 25 °C for 20 min before being centrifuged at 16,873 $g$ for 15 min at 4 °C. The supernatants were collected for determination of acetyl-CoA concentration using an acetyl-CoA assay kit (Ji Ning Biotech, JN709212). The nuclei were extracted as previously[46] reported. In general, rice tissues were ground with liquid nitrogen and mixed with PBS buffer containing a protease inhibitor cocktail (Roche, 11697498001). The homogenate was filtered through Miracloth (Millipore, 475855). The extract was centrifuged at 1500 $g$ at

4 °C for 10 min. The resulting pellet was the nuclear fraction, while the supernatant was further centrifuged at 10,000 $g$ at 4 °C for 10 min to obtain the soluble cytoplasmic fraction. The nuclear acetyl-CoA content was assayed.

The content of glucose and citrate was measured as follows: 0.1 g of 7-day-old rice green tissues were crushed. The cytosol and nuclei fractions were extracted using the method described above. The glucose or citrate content of each fraction was assayed using the Plant Glucose ELISA Kit (Ji Ning Biotech, JN732302) or the Plant Citrate Acid ELISA Kit (Ji Ning Biotech, JN737021), respectively.

## RT-qPCR analysis

TRIzol reagent (Invitrogen, 15596018) was used to isolate total RNA. The reverse transcription kit (Vazyme, R212-01) was used to create complementary DNA from 1 μg of RNA. SYBR Premix ExTaq (TaKaRa, RR820A) real-time PCR was done on an ABI 7500 real-time PCR equipment. Rice ACTIN was utilized as an internal standard for normalizing gene expression. Triplicate reactions were performed for each sample. Relative expression was measured using $2^{-\triangle\triangle CT}$ method[47]. Primers used in RT-qPCR were listed in Supplementary Data 5.

## Protein structural modeling and phylogenetic tree construction

The structure of human ACL (PDB ID:6HXH)[48] was downloaded from the PDB protein data bank (https://www.pdbus.org/). The structure of rice ACL was generated by AlphaFold Protein Structure Database (https://www.alphafold.ebi.ac.uk/). The structural overlap of human and rice ACL was generated by PyMOL (Version 2.5.2). For computational modeling of the protein interaction between GNATs and ACLA2, the 3-D structures of GNATs and ACLA2 were downloaded from the AlphaFold Protein Structure Database (https://www.alphafold.ebi.ac.uk/) and analyzed by ZDOCK (http://zdock.umassmed.edu/) for primary docking. Interacting amino acids were analyzed by PDBePISA (https://www.ebi.ac.uk/msd-srv/prot_int/pistart.html). Structural figures were generated by PyMOL (Version 2.5.2).

The proteins from human and Arabidopsis were aligned with ClustalX along with the rice GNAT proteins identified in this study. The MEGA (v5.2.2) program was used to conduct the phylogenetic analysis. The Neighbor-Joining method was used to estimate evolutionary distances.

## Protein–protein interaction assay

For yeast two-hybrid analysis, rice HATs and ACL cDNAs were cloned into the plasmids pGBKT7 and pGADT7 (Clontech), respectively. Yeasts containing the pGBKT7 and pGADT7 plasmids were grown on SD-LTHA (-Leu/-Trp/-His/-Ade) medium (solid) to select interaction-positive clones.

For in vitro pull-down assay, His-tagged HAG704 and GST-tagged ACLA2 were constructed and expressed in the *Escherichia coli* cells. GST or ACLA2-GST coupled GST beads (GE Healthcare, 17–5132-01) were used to pull down HAG704-6×His and an anti-His antibody (1:1000, Abcam, ab213204) was applied to detect HAG704-6×His. Similarly, His or HAG704-His coupled Ni-NTA Magnetic Agarose beads (QIAGEN, 36113) were used to pull down ACLA2-GST and an anti-GST antibody (1:1000, Abcam, ab19256) was applied to detect ACLA2-GST. The original raw blotting images are shown as a Source Data file.

For the BiFC test, *A. tumefaciens* cells were transformed with split luciferase nLUC- and cLUC and subsequently infiltrated into 6-week-old tobacco (*Nicotiana benthamiana*) leaves. After 48 h of transfection, tobacco leaves were infiltrated with 1 mM luciferin (Gold Biotechnology, 115144-35-9). Luciferase bioluminescence images were taken with the Chemi-Image System (Tanon 5200Multi, China).

For co-immunoprecipitation in rice protoplasts, the proteins were extracted from rice protoplasts expressing GFP or FLAG-tagged constructs using lysis buffer (10 mM Tris/HCl pH 7.5, 150 mM NaCl, 0.5 mM

EDTA, 0.5% Nonidet™ P40 Substitute) and then incubated with anti-GFP agarose beads (ChromoTek, gtma) or anti-HAG704 (the KQKGTDAAADPKKRR and DADDQTVEQQPEDLKTQE sequences of HAG704 protein was used as mixed antigen to produce polyclonal antibodies in rabbit) coated with protein-A magnetic beads (Thermo Fisher Scientific, 10001D) for 5 h at 4 °C. After four washes with washing/dilution buffer (10 mM Tris/Cl pH 7.5, 150 mM NaCl, 0.5 mM EDTA), the co-immunoprecipitated proteins were separated by SDS-PAGE and detected with anti-GFP (1:1000, Abcam, ab290), anti-HAG704 (1:1000, homemade), and anti-FLAG (1:1000, Sigma, F3165) antibodies. The original raw blotting images are shown as a Source Data file.

For co-immunoprecipitation in wild type rice tissues (calli or root tip) or 3 DAF HAG704 OE seeds, the samples were ground into power by liquid nitrogen and extracted with lysis buffer (10 mM Tris–HCl pH 7.5, 150 mM NaCl, 0.5 mM EDTA, 0.5% Nonidet™ P40 Substitute) and then incubated with Anti-FLAG M2 magnetic beads (M8823, Sigma) or anti-HAG704 coated with protein-A magnetic beads (Thermo Fisher Scientific, 10001D) for 5 h at 4 °C. After four washes with PBST buffer, the co-immunoprecipitated proteins were separated by SDS-PAGE and detected with anti-FLAG (1:1000, Sigma, F3165), anti-ACLA2 (1:1000, the full length ACLA2 protein was used as antigen to produce polyclonal antibodies in rabbit), and anti-HAG704 (1:1000, homemade) antibodies. The original raw blotting images are shown as a Source Data file.

### In vitro histone acetyltransferase activity assay

For in vitro histone acetyltransferase activity assay, vectors of HAG704, ACLA2, ACLB-3×FLAG, and H4-6×His were transformed into tobacco (*Nicotiana benthamiana*) leaf epidermal cells. 48 h after the transfection, the transfected tobacco leaves were collected and ground into powder by liquid nitrogen. Then the proteins were extracted with lysis buffer (10 mM Tris-HCl pH 7.5, 150 mM NaCl, 0.5 mM EDTA, 0.5% Nonidet™ P40 Substitute) and purified with Ni-NTA Magnetic Agarose beads (QIAGEN, 36113). The extracted proteins were boiled at 90 °C for 10 min and then analyzed using Western blotting. The primary antibodies used were anti-HAG704 (1:1000, in house), anti-H3 (1:1000, Abcam, ab1791), anti-H3K9ac (1:1000, Millipore, 07–352), anti-H3Kac (1:1000, Millipore, 17-615), anti-H4 (1:1000, Abcam, ab177840), anti-H4K12ac (1:1000, PTM-Biolab, PTM-121), anti-H4K5ac (1:1000, Millipore, 07-327), anti-H4K16ac (1:1000, Millipore, 07-329), anti-ACLA2 (1:1000, homemade), anti-FLAG (1:1000, Sigma, F3165). The secondary antibodies used were peroxidase-conjugated goat anti-rabbit antibody (1:10000, Abbkine, A21020), and peroxidase-conjugated goat anti-mouse antibody (1:10000, Abbkine, A21010). The original raw blotting images are shown as a Source Data file.

### Histone extraction and western blot

The EpiQuik Total Histone Extraction Kit (Epigentek USA, OP-0006-100) was used to extract total histone-enriched fractions from 6 DAF rice seeds or seven-day-old rice roots. For cytosolic histone extraction, the previous reported protocol was followed with modifications[49]. First, 5–10 g 7-day-old rice roots were lysed into powder in the liquid nitrogen and dissolved in lysis buffer (10 mM Tris-HCl, pH 8.0, 0.4 M sucrose, 0.1 mM phenylmethylsulfonyl fluoride [PMSF], 10 mM MgCl2, 5 mM β-mercaptoethanol, 1 protease inhibitor cocktail). Next, the mixture was cleaned by filtering through a double layer of Miracloth (Millipore, 475855) and centrifuged at 1,000 g at 4 °C for 20 min. The supernatant was kept for cytosolic histones by acid extraction. The chromatin precipitation was washed at least five times with washing buffer (0.25 M sucrose, 10 mM Tris, pH 8.0, 1% Triton X-100, 10 mM MgCl2, 0.1 mM PMSF, 5 mM β-mercaptoethanol, 1 protease inhibitor cocktail) and kept as the nucleosomal histones. The extracted histone proteins were analyzed using immunoblot analysis, and the following antibodies were used: anti-H3K9ac (1:1000, Millipore, 07-352), anti-

H3K14ac (1:1000, Abcam, ab52946), anti-H3K18ac (1:1000, Millipore, 07-354), anti-H3K27ac (1:1000, Abcam, ab4729), anti-H3 (1:1000, Abcam, ab1791), anti-H3Kac (1:1000, Millipore, 17-615), anti-H4 (1:1000, Abcam, ab177840), anti-H4K5ac (1:1000, Millipore, 07-327), anti-H4K12ac (1:1000, PTM-Biolab, PTM-121), anti-H4K16ac (1:1000, Millipore, 07-329), anti-H4Kac (1:1000, Millipore, 06-598), and peroxidase-conjugated goat anti-rabbit antibody (1:10000, Abbkine, A21020). Immunoblotting results were quantified by ImageJ (v1.6.0_24). The original raw blotting images are shown as a Source Data file.

### Immunocytochemistry

Paraffin sections from each sample were first cut onto slides, dewaxed in xylene, dehydrated in graded ethanol, and microwaved at high intensity in citrate buffer (pH 6.0). Then, the sections were treated with 3% $H_2O_2$ in PBS for 10–30 min to block endogenous peroxidase, followed by blocking with 2–5% normal bovine serum albumin in PBS (pH 7.4) for 1 h. For the endosperm nuclei or root cells/nuclei, they were first spread onto the slides directly and then allowed to dry at room temperature. Next, all sections were incubated with primary antibodies (anti-HAG704 or anti-ACLA2, homemade) diluted 1:100 and the secondary antibody (Invitrogen, A-11011) diluted 1:200 in PBST (pH 7.4, PBS with 0.1% Tween 20) at room temperature for 2 h, respectively. After three washes with PBST, the sections were stained with DAPI and viewed with a Nikon Ni-E light microscope equipped with a CCD camera.

### ChIP-qPCR

0.5–1 g of rice seeds (3 DAF) or leaves (30-day-old) was vacuum-crosslinked in 1% formaldehyde. Ultrasound fragmented chromatin to 200–800 bp. 5 μL anti-H4K5ac (Millipore, 07-327) or anti-HAG704 and 40 μL protein A beads (Invitrogen, 88845) were incubated at 4 °C for more than 4 h. About 100 μL fragmented chromatin suspension was added after bead washing, then incubated overnight at 4 °C. After washing, reverse cross-linked immunoprecipitated chromatin was used for qPCR. ChIP-qPCR primers were listed in Supplementary Data 5. Three biological replicates were performed using samples collected from three independent cultures.

### RNA-seq data analysis

RNA was isolated using TRIzol reagent (Invitrogen, 15596026). The RNA-seq libraries were produced with the Illumina TruSeq RNA Sample Preparation Kit and sequenced on the Illumina HiSeq 2000. RNA-seq data was filtered by FastP (v0.232) to remove contamination and low-quality reads. FeatureCounts (version 2.0.3) and DESeq2 (v1.36.0) were used to map clean reads to rice genome (MSU 7.0) and calculate differentially expressed genes, respectively. Genes with $p$ value < 0.05 and fold change > 2 in mutants were considered differentially expressed. ESGs and EnSGs were generated by TissueEnrich software[30].

### CUT&Tag, sequencing, and analysis

CUT&Tag experiments were performed as described previously[32] with minor modifications. $3 \times 10^5$ endosperm free nuclei were harvested, washed with wash buffer (20 mM HEPES pH 7.5, 150 mM NaCl, 0.5 mM spermidine), and immobilized to concanavalin A-coated beads (Vazyme, TD902) with incubation at room temperature for 10 min. The bead-bound nuclei were incubated in 200 μl of primary antibody buffer (wash buffer with 1% BSA, 2 mM EDTA and 0.05% digitonin for gentle permeabilization of the nuclear membrane) with 1:50 anti-H4K5ac (Millipore, 07-327) or anti-H4K16ac (Millipore, 07-329) antibody dilution at 4 °C by rotating overnight. The next day, the primary antibody buffer was removed and cells were washed with 800 μl of dig-wash buffer (wash buffer with 1% BSA and 0.05% digitonin) three times. After washing, antibody-incubated nuclei were resuspended in 200 μl of dig-wash buffer with a 1:100 dilution of secondary antibody (Vazyme, TD902) and incubated at room temperature with gentle

rotation for 1 h. To eliminate binding antibodies, nuclei were washed three times with 800 µl dig-wash buffer. After a brief wash with dig-wash buffer, nuclei were resuspended in 200 µl of dig-300 buffer (20 mM HEPES pH 7.5, 300 mM NaCl, 0.5 mM spermidine, 1% BSA and 0.01 % digitonin) and incubated at room temperature for 1 h with gentle rotation. pA-Tn5-bound nuclei were washed with 800 µl of dig-300 buffer three times, then tagmented at 37 °C for 1 h. 15 mM EDTA, 500 µg/ml proteinase K, and 0.1% SDS were added after tagmentation to terminate tagmentation and degrade protein. Genomic DNA was extracted and purified with VAHTS DNA Clean Beads (Vazyme, N411-01). Purified genomic DNA was amplified using the universal i5 primer and barcoded i7 primer (Vazyme, TD202) using Vazyme Master Mix (Vazyme, TD902). The library PCR products were cleaned with Agencourt AMPure XP beads (Beckman Coulter, A63881) and sequenced on Illumina HiSeq 2000. FastP (v0.232) was used to remove low-quality reads and to trim low-quality bases as well as adapters, with default parameters. Trimmed reads were aligned to the rice genome (MSU7.0) by Bowtie2 (version 2.3.5.1) with default settings[50]. Then, duplicated reads were discarded using samtools (v1.9)[51]. MACS software (version 2.2.7.1)[52] was used to call histone modification peaks by default parameters (-f BAMPE -B -q 0.05). Wig files produced by deepTools (v2.5.3)[53] software with BPM normalization were used for data visualization by IGV (version 2.3.88). DiffBind (v3.5)[54] with default parameters was used to identify differential histone modification sites between mutants and wild types using two biological replicate BAM files from each group (e.g., WT and mutant) as input. Annotation of peaks was performed using homer (v4.11) annotatePeaks.pl with default parameters[55]. String database (v2.0)[56] was used to conduct the gene ontology analysis. Heatmaps were generated by Tbtools (v0.6)[57]. Ridge plots and scatter plots were generated in R (v3.5). For the scatter plots, the strength of the relationship between variables is indicated by the Pearson's correlation coefficient ($r$). The calculation of Pearson's $r$ was performed using the "cor()" function in the R (v3.5).

## Statistics and reproducibility
The data are presented as mean ± standard deviation (SD). Statistical analysis was performed using Student's t-test (paired, two-sided) and Wald test (unpaired, two-sided) for comparing two groups, and one-way ANOVA with Tukey's multiple comparison tests for comparing three or more groups. $P < 0.05$ was considered as statistically significant. Data analysis was conducted using Microsoft Excel (version 2304) and GraphPad Prism Software (Version 8.0.1). Detailed statistical information can be found in the figure legends.

The RNA-seq data generated in-house consisted of three biological replicates for each genotype, and the CUT&Tag data comprised two biological replicates for each genotype. The biological replicates of all other experiments presented in this study are indicated in the respective figure legends. No statistical method was used to predetermine sample size. No data were excluded from the analyses. The experiments were not randomized. The Investigators were not blinded to allocation during experiments and outcome assessment.

## Reporting summary
Further information on research design is available in the Nature Portfolio Reporting Summary linked to this article.

## Data availability
The in-house RNA-seq data and CUT&Tag data generated in this study were deposited to the NCBI SRA database under accession code PRJNA874160. Other previously published RNA-seq[58–61] data used in this study are available in the NCBI SRA database under accession codes PRJNA13141, PRJNA262032, SRP008821, and in the DDBJ database under the accession codes DRA009458 and DRA007969. The structure of human ACL was downloaded from the PDB database under the accession number 6HXH. Extra data for the individual measurements are available on request. The source data for Figs. 1a, d-g, 2a, 5d, 6b, c, 7c, and Supplementary Figs. 3a, 10b, 11 are provided in a Source Data file. This file also includes uncropped and unprocessed scans of the western blots for Figs. 2b, c, 3d, f, g, 4, and Supplementary Figs. 2f, 3b, 6. Source data are provided with this paper.

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

## Acknowledgements

We thank Qinglu Zhang, Prof. Xianghua Li, and Prof. Lizhong Xiong for assistance. We also thank for the support of BaiChuan Fellowship of College of Life Science and Technology, Huazhong Agricultural University. This work was supported by grants from the National Natural Science Foundation of China (No. 31821005 to Y.Z., 32100465 to Q.X., 31730049 and 32070563 to D.X.Z.), China Postdoctoral Science Foundation (No. 2021M691183 to Q.X.), Fundamental Research Funds for the Central Universities (2662023SKPY002 to Y.Z.), and the French Agence Natonale de la Recherche (ANR-19-CE12-0027-01 to D.X.Z.).

## Author contributions

Q.X. and Y.Y. performed most of the experiments and analyzed the data. B.L., Z.C., and Y.Z. performed western blot and BIFC experiments. X.M. contributed to the creation of double mutant plants. J.W. helped with the endosperm free nuclei collection. D.X.Z. supervised the project, analyzed the data, and wrote the paper with inputs from Q.X. and Y.Y. All authors read and approved the final manuscript.

## Competing interests

The authors declare no competing interests.
