## [Peer Review File · Nature Communications]

REVIEWER COMMENTS

Reviewer #1 (Remarks to the Author):

The manuscript by Xu et al., examined the role of ATP-citrate lyase subunits 2 (ACLA2) and histone acetyltransferase HAG703 in the regulation of H4K5 acetylation and cell proliferation. They showed that the two proteins interact in the nucleus to regulate histone H4K5 acetylation at largely the same genomic regions. The losses of ACLA2 and HAG703 lead to S phase stagnation and the mis-expression of similar sets of genes. These data demonstrate that the HAG703-ACLA2 module selectively promotes histone acetylation at specific lysine residues, required for cell division. I believe these are wonderful pieces of data that examined an important question of how the energy metabolism could cope with cell division through the bridge of acetyl-CoA and histone acetylation. However, I do have several concerns that the authors need to address.

Major points:

1. Figure S1c and d, the subcellular localization of ACLA2 and ACLB was found in the nucleus and cytoplasm. However, the strong 35S promoter was used to drive the two genes. The authors should use native promoter-driven rice lines to indicate that the two proteins are indeed nucleocytoplasmic localized.
2. Line 147-149, there are four families of histone acetyltransferases, including (CBP)/p300, the TAF, the MYST and GNAT subfamilies. However, the authors only chose GNAT family proteins to test how ACLAs interact with HATs. The authors should test or exclude that ACLAs do not interact with the other three histone acetyltransferase families.
3. The phenotype of *acla2* mutant is weaker than *hag704* mutant. However, the ACLA2 also interacts with HAG702 and HAG703 in yeast cells. It is possible that ACLA2 is also required for the HAG702/3 function. Thus, the loss of *acla2* should result in loss of function of all three HAG702/3/4 and more severe phenotypes than *hag704* single mutant. The functional interaction between ACLAs and HAG702/3/4 is therefore much more complicate than current proposed. The authors need to revise this part of conclusion.
4. Figure 3, the HAG704 and ACLA2 interaction. Y2H, BiFC, and Co-IP using highly-overexpressed HAG704 and ACLA2 in tobacco leaves are not convincing. The Co-IP was also performed in rice tissues in figure 3g, but it seems to use overexpressed HAG704-FLAG driven by ubiquitin promoter (methods). Thus, proper co-IP assays need to be conducted using rice lines expressing functional tagged HAG driven by the native promoter in mutant background.
5. The authors claim that HAG704 and ACLA2 are in the same pathways to control root development. However, the root length of the *hag704 acla2* double mutant is obviously shorter than that of the *hag704* single mutant, suggesting that the two proteins function in different pathways. Consistent with this interpretation, HAG704 is required for H4K5 and H4K16 acetylation, while ACLA2 is required for

H3K14, H3K27, H3K9 and H4K5 acetylation. Thus, HAG704 and ACLA2 only share one common lysine residue (H4K5) for acetylation, but differ at other lysine residues. The authors should revise their conclusion.

Minor points:

1. In the title, the authors claim that H4K5 acetylation is required for DNA replication and cell proliferation. However, the *acla2* and *hag704* mutants showed reduced H3K14 and H4K16 acetylation, respectively. It is possible that the reduction of these two histone acetylations also contributed to the decreased cell proliferation. Thus, it is better to change the second "to" into "and".
2. The authors only examined the function of ACLA2 in H4K5 acetylation among three ACLAs. It is currently unknown whether ACLA1 and ACLA3 are involved in regulating H4K5ac. Thus, ATP-citrate lyase in the title and ACL throughout the manuscript (for example, Lines 142-144) should be replaced by ATP-citrate lyase subunits 2 (ACLA2).
3. Is the anti-HAG704 antibody homemade or commercial? The methods producing the antibody should be shown in materials and methods if homemade.

Reviewer #2 (Remarks to the Author):

The manuscript by Qitao Xu et al. reports rice ACL cooperates with HAT1 to regulate plant growth by mediating histone acetylation, mainly at H4K5ac. The study is well-performed, well-written and the data is nicely presented. Although there are still several points that need to further clarify and carefully analyzed, in general, this work provides a direct link that how the metabolite changes are sensed by epigenetic modification in plants. My detailed comments are shown as follows:

1. To me, the phenotypes of *hag704-1/acla2-1* shown in Figure S2g are worse than *hag704* in terms of shorter root length and more brown lesions, indicating additive phenotypes in double mutants compared with every single mutant. Plus, the changing patterns of histone acetylation are different in *hag704* (mainly on H4) and *acla2* (on both H3 and H4). Therefore, the authors may tone down their claim on "... HAG704 and ACLA2 function in the same pathways to control seed and root meristem development."
2. In mammalian cells, several enzymes in TCA cycles can be found in the nucleus, suggesting an independent pool of TCA-related metabolites. In Figure 2a, the authors detected the different amounts of acetyl-CoA in the nucleus and entire cells, suggesting the relative independence of metabolite pools in the nucleus and cytosol. Therefore, a key question is how the energy state of an entire cell has been sensed in the nucleus. The authors may quantify some upstream metabolites, such as citrate (the substrate of ACL) and glucose in the nucleus and cytosol, to further illustrate the independence of metabolite pools in plants. Or, at least, the authors need to discuss this point.

3. As shown in Figure S4b, histone acetylation in the nucleus rather than cytosol was affected in the hag704 mutant. Both H4 signals were detected in two compartments, thus the authors need to include another marker protein, such as actin, to justify the effectiveness of cell fractions.
4. Following the above question, the authors need to mention all other immunoblots (histone acetylation) in different figures were generated by using total protein extracts or just nuclei. As ACLA2 and ACLB can be detected in cytosol, I am wondering the decreased histone acetylation may be detected both in cytosol and nucleus after knocking out acla2.
5. The authors applied Person correlation to show the relationship between two conditions. However, the plotting regions were mentioned neither in methods nor figure legends, for example, in Figure 5d, changes on total rice genes or H4K5ac bound genes in specific genetic backgrounds. A similar case is in Figure S8b.
6. As the authors performed two biological replicates CUT&Tag for H4K5ac and H4K16ac, they did not mention how they handled the replicates to generate the following data. (One replicate per sample or generating average value from two replicates)
7. The authors compared the histone acetylation changes by CUT&Tag. However, they did not provide the reads information, such as raw reads, and uniquely mapped reads for each sample. Without the information, it is hard to conclude whether the observed differences result from biological factors or sequencing depth. Also, is this a spiked analysis? Different IP efficiency may affect sequencing results.

Reviewer #1 (Remarks to the Author):

The manuscript by Xu et al., examined the role of ATP-citrate lyase subunits 2 (ACLA2) and histone acetyltransferase HAG703 in the regulation of H4K5 acetylation and cell proliferation. They showed that the two proteins interact in the nucleus to regulate histone H4K5 acetylation at largely the same genomic regions. The losses of ACLA2 and HAG703 lead to S phase stagnation and the mis-expression of similar sets of genes. These data demonstrate that the HAG703-ACLA2 module selectively promotes histone acetylation at specific lysine residues, required for cell division. I believe these are wonderful pieces of data that examined an important question of how the energy metabolism could cope with cell division through the bridge of acetyl-CoA and histone acetylation. However, I do have several concerns that the authors need to address.

Major points:

1. Figure S1c and d, the subcellular localization of ACLA2 and ACLB was found in the nucleus and cytoplasm. However, the strong 35S promoter was used to drive the two genes. The authors should use native promoter-driven rice lines to indicate that the two proteins are indeed nucleocytoplasmic localized.

Thank you for the comments.

We have constructed a native ACLB promoter-CDS-GFP fusion construct and transformed this construct into wild-type rice protoplasts to observe its subcellular localization. The results shown in Supplementary Fig. 1d in this revised version indicate that ACLB was localized in both the nucleus and cytoplasm of rice cells.

For ACLA2, since we had produced the antibody, we have performed immunostaining of wild type rice root tip cells. As shown in Supplementary Fig. 1d, ACLA2 was also detected in both the nucleus and cytoplasm of rice cells, consistent with the GFP results.

2. Line 147-149, there are four families of histone acetyltransferases, including (CBP)/p300, the TAF, the MYST and GNAT subfamilies. However, the authors only chose GNAT family proteins to test how ACLAs interact with HATs. The authors should test or exclude that ACLAs do not interact with the other three histone acetyltransferase families.

Thanks for the suggestion.

As suggested, we have tested ACLA2 interaction with other HAT family members (CBP/P300, TAF and MYST) with Y2H assays. We found that ACLA2 could also interact with CBP/P300 and TAF1 but not MYST members.

These results are added in Supplementary Fig. 4, and are discussed in text.

3. The phenotype of *acla2* mutant is weaker than *hag704* mutant. However, the ACLA2 also interacts with HAG702 and HAG703 in yeast cells. It is possible that ACLA2 is also required for the HAG702/3 function. Thus, the loss of *acla2* should result in loss of function of all three HAG702/3/4 and more severe phenotypes than *hag704* single mutant. The functional interaction between ACLAs and HAG702/3/4 is therefore much more complicated than currently proposed. The authors need to revise this part of conclusion.

Thanks for the comments. We agree that the *acla2* phenotype is more pleiotropic than *hag704*. We have revised the related statements/conclusions in the MS.

4. Figure 3, the HAG704 and ACLA2 interaction. Y2H, BiFC, and Co-IP using highly-overexpressed HAG704 and ACLA2 in tobacco leaves are not convincing. The Co-IP was also performed in rice tissues in figure 3g, but it seems to use overexpressed HAG704-FLAG driven by ubiquitin promoter (methods). Thus, proper co-IP assays need to be conducted using rice lines expressing functional tagged HAG driven by the native promoter in mutant background.

We apologize for the confusing presentation of Fig. 3g.

Indeed, the results shown in Fig. 3g (middle, previously right) were from the Co-IP experiments with the antibodies of HAG704 and ACLA2 proteins to analyze wild type rice cells (callus). During the revision, we have performed similar Co-IP experiments with the same antibodies to analyze wild type rice root tissues (tips) and have obtained the same results shown in the revised Fig. 3g (right panel).

5. The authors claim that HAG704 and ACLA2 are in the same pathways to control root development. However, the root length of the *hag704 acla2* double mutant is obviously shorter than that of the *hag704* single mutant, suggesting that the two proteins function in different pathways. Consistent with this interpretation, HAG704 is required for H4K5 and H4K16 acetylation, while ACLA2 is required for H3K14, H3K27, H3K9 and H4K5 acetylation. Thus,

HAG704 and ACLA2 only share one common lysine residue (H4K5) for acetylation, but differ at other lysine residues. The authors should revise their conclusion.

Thank you for the point. We agree with the comments. We have revised the related statement/discussion.

Minor points:

1. In the title, the authors claim that H4K5 acetylation is required for DNA replication and cell proliferation. However, the *acla2* and *hag704* mutants showed reduced H3K14 and H4K16 acetylation, respectively. It is possible that the reduction of these two histone acetylations also contributed to the decreased cell proliferation. Thus, it is better to change the second "to" into "and".

Thanks for the suggestion. We have revised our titles as suggested.

2. The authors only examined the function of ACLA2 in H4K5 acetylation among three ACLAs. It is currently unknown whether ACLA1 and ACLA3 are involved in regulating H4K5ac. Thus, ATP-citrate lyase in the title and ACL throughout the manuscript (for example, Lines 142-144) should be replaced by ATP-citrate lyase subunits 2 (ACLA2).

Thanks for the suggestion.

We have changed ACL to ACLA2, as suggested.

3. Is the anti-HAG704 antibody homemade or commercial? The methods producing the antibody should be shown in materials and methods if homemade.

The anti-HAG704 antibody was produced in-house and we have added this information to the Methods section.

Reviewer #2 (Remarks to the Author):

The manuscript by Qiutao Xu et al. reports rice ACL cooperates with HAT1 to regulate plant growth by mediating histone acetylation, mainly at H4K5ac. The study is well-performed, well-written and the data is nicely presented. Although

there are still several points that need to further clarify and carefully analyzed, in general, this work provides a direct link that how the metabolite changes are sensed by epigenetic modification in plants. My detailed comments are shown as follows:

1. To me, the phenotypes of *hag704-1/acla2-1* shown in Figure S2g are worse than *hag704* in terms of shorter root length and more brown lesions, indicating additive phenotypes in double mutants compared with every single mutant. Plus, the changing patterns of histone acetylation are different in *hag704* (mainly on H4) and *acla2* (on both H3 and H4). Therefore, the authors may tone down their claim on "... HAG704 and ACLA2 function in the same pathways to control seed and root meristem development."

Thanks for the point. We have revised the statement/discussion relevant to the double mutants in the MS text.

2. In mammalian cells, several enzymes in TCA cycles can be found in the nucleus, suggesting an independent pool of TCA-related metabolites. In Figure 2a, the authors detected the different amounts of acetyl-CoA in the nucleus and entire cells, suggesting the relative independence of metabolite pools in the nucleus and cytosol. Therefore, a key question is how the energy state of an entire cell has been sensed in the nucleus. The authors may quantify some upstream metabolites, such as citrate (the substrate of ACL) and glucose in the nucleus and cytosol, to further illustrate the independence of metabolite pools in plants. Or, at least, the authors need to discuss this point.

Thanks for the suggestion. We have measured citrate and glucose contents in the cytoplasmic and nuclear fractions of rice cells. We have observed similar levels of the metabolites in both fractions, supporting an independent glycolysis/TCA pool in the nucleus. Interestingly, we detected higher citrate levels in both the cytoplasmic and nuclear fractions and higher glucose levels in the cytoplasm of *acla2* plants than the wild type, which would suggest that the mutation might lead to the metabolic accumulation. These results are added in Supplementary Fig. 3a and are discussed in the text.

3. As shown in Figure S4b, histone acetylation in the nucleus rather than cytosol was affected in the *hag704* mutant. Both H4 signals were detected in two compartments, thus the authors need to include another marker protein, such

as actin, to justify the effectiveness of cell fractions.

Thank you for your suggestion. We have repeated the experiments by first testing the presence or absence of actin in the cytoplasmic and nuclear fractions before performing acetylation immunoblots. The histone acetylation results shown in Supplementary Fig.6b are similar to the initial data presented in the previous version.

4. Following the above question, the authors need to mention all other immunoblots (histone acetylation) in different figures were generated by using total protein extracts or just nuclei. As *ACLA2* and *ACLB* can be detected in cytosol, I am wondering the decreased histone acetylation may be detected both in cytosol and nucleus after knocking out *acla2*.

Thanks for the suggestion.

We have indicated that the total histone extracts were used in the figure legends of Fig. 2, 4 and Supplementary Fig. 6.

As suggested, we have tested histone acetylation levels in the cytosol and nuclear fractions of *acla2*. Decreases of histone acetylation were detected only in the nucleus (please see Supplementary Fig. 3b). The result is commented in text.

5. The authors applied Person correlation to show the relationship between two conditions. However, the plotting regions were mentioned neither in methods nor figure legends, for example, in Figure 5d, changes on total rice genes or H4K5ac bound genes in specific genetic backgrounds. A similar case is in Figure S8b.

Thanks for the suggestion.

The histone acetylation CUT&Tag peaks were used for plotting. In Fig. 5d, the correlation analysis was based on the differential H4K5ac peaks commonly detected in *acl2* vs WT and *hag704* vs WT (N=26,995). This is now explained in Fig. 5d legend.

In Supplementary Fig. 10b (previous version was Figure S8b) the correlation between of the differential H4K5ac and H4K16ac peaks (N=18,749) in *hag704* vs WT was analyzed. This is now explained in Supplementary Fig. 10b legend.

6. As the authors performed two biological replicates CUT&Tag for H4K5ac and H4K16ac, they did not mention how they handled the replicates to generate the following data. (One replicate per sample or generating average value from two replicates).

Thank you for the suggestion. We used the software "DiffBind" (Stark & Brown, 2011, cited in the Methods) to analyze the replicates. This program produces mean normalized reads from the input replicates for each group (e.g., mutant or WT) as well as for fold change and significance analysis.

7. The authors compared the histone acetylation changes by CUT&Tag. However, they did not provide the reads information, such as raw reads, and uniquely mapped reads for each sample. Without the information, it is hard to conclude whether the observed differences result from biological factors or sequencing depth. Also, is this a spiked analysis? Different IP efficiency may affect sequencing results.

Thanks for the suggestion.

We have provided the raw sequence reads, mapping statistics, including total and unique mapping reads in Supplementary Table 2.

The mapping rates of clean reads were >91% for both H4K5ac and H4K16ac datasets. Unique mapping rates were >61% for H4K5ac and >71% for H4K16ac datasets. These unique mapping rates exceed the recommended threshold of 50% for ChIP-seq analysis (Bailey et al., 2013).

Regarding the spike-in, we did not use it in our data analysis. One reason for this was that the normalization method we used in DiffBind requires the raw read counts as input, and spike-ins are not necessary for this approach. We also consulted with Steven Henikoff's group, experts in CUT&Tag data analysis, who confirmed that spike-in is recommended but may not be essential and recommended using DESeq2 (used in DiffBind) for data normalization, which we followed. ChIP-qPCR experiments validated the reliability of our CUT&Tag data analysis.

Reference

Bailey T, Krajewski P, Ladunga I, Lefebvre C, Li Q, Liu T, Madrigal P, Taslim C, Zhang J. 2013. Practical guidelines for the comprehensive analysis of ChIP-seq data. PLoS Comput Biol 9(11): e1003326.

REVIEWERS' COMMENTS

Reviewer #1 (Remarks to the Author):

The authors have addressed my comments appropriately. Thanks.

Reviewer #2 (Remarks to the Author):

Generally, the current manuscript has addressed most of my concerns mentioned previously, except the last point referring to normalization and Spik-in methods.

In fact, similar to many other normalizing methods, DESeq2 is based on the assumption that the occupancy of the protein of interest remains unaltered at the majority of sites (ChIP-seq) under the various conditions studied¹. The mutation of either HAG704 or ACLA2 presumably causes the global reduction of acetylated histone, which is not fit with the assumption mentioned above. In addition, the spike-in reads in each sample can be easily extracted from total reads and can be used to adjust sample reads across different conditions, thus I am not clear why the spike-in approach is conflict with the usage of DiffBind.

Multiple research pointed out the potential flaws of conventional normalization in dealing with histone modifications in the mutants/inhibitor of key enzymes, such as EZH2 to H3K27me³², and DOT1L to H3K79me²³. Thus, the authors may carefully analyze their genome-wide data and draw conclusions.

Minor points:

1, typos in line 453 "ACAL2" (twice)

2, In Table S1, the format of numbers in Total raw reads is inconsistent with other reads numbers.

1 Greulich, F., Mechtidou, A., Horn, T. & Uhlenhaut, N. H. Protocol for using heterologous spike-ins to normalize for technical variation in chromatin immunoprecipitation. STAR Protocols 2, 100609, doi:10.1016/j.xpro.2021.100609 (2021).

2 Bonhoure, N. et al. Quantifying ChIP-seq data: a spiking method providing an internal reference for sample-to-sample normalization. Genome Research 24, 1157-1168, doi:10.1101/gr.168260.113 (2014).

3 Chen, K. et al. The Overlooked Fact: Fundamental Need for Spike-In Control for Virtually All Genome-Wide Analyses. Molecular and Cellular Biology 36, 662-667, doi:10.1128/MCB.00970-14 (2015).

Reviewer #1 (Remarks to the Author):

The authors have addressed my comments appropriately. Thanks.

Response:

Thanks for the reviewer's insightful suggestions and comments, which have greatly helped us further improve the manuscript.

Reviewer #2 (Remarks to the Author):

Generally, the current manuscript has addressed most of my concerns mentioned previously, except the last point referring to normalization and Spik-in methods.

In fact, similar to many other normalizing methods, DESeq2 is based on the assumption that the occupancy of the protein of interest remains unaltered at the majority of sites (ChIP-seq) under the various conditions studied¹. The mutation of either HAG704 or ACLA2 presumably causes the global reduction of acetylated histone, which is not fit with the assumption mentioned above. In addition, the spike-in reads in each sample can be easily extracted from total reads and can be used to adjust sample reads across different conditions, thus I am not clear why the spike-in approach is conflict with the usage of DiffBind. Multiple research pointed out the potential flaws of conventional normalization in dealing with histone modifications in the mutants/inhibitor of key enzymes, such as EZH2 to H3K27me3², and DOT1L to H3K79me3³. Thus, the authors may carefully analyze their genome-wide data and draw conclusions.

Response:

Thank you for your valuable feedback and suggestions.

We acknowledge that DESeq2 assumes the occupancy of the protein of interest remains unaltered at the majority of sites. In fact, in the *hag704* and *acla2* mutants, about 10% (H4K16ac) to 20% (H4K5ac) of total peaks showed reduced histone acetylation. This indicates that a substantial portion of the genome remains unaffected. It seems to us using DESeq2 is appropriate for the analysis.

We agree that with the spike-ins approach it would be better for such data analysis. Unfortunately, spike-ins controls were not included in the samples for high throughput sequencing (which seemed not essential according to Henikoff's group). However, we have addressed this issue by performing ChIP-qPCRs to validate the CUT&Tag results.

Minor points:

1, typos in line 453 "ACAL2" (twice)

Response

Thanks. We have corrected that (Line 460).

2, In Table S1, the format of numbers in Total raw reads is inconsistent with other reads numbers.

Thanks for pointing this out. We have corrected that. The content in Table S1 has been placed in Supplementary Data 1 in this revision.

1 Greulich, F., Mechtidou, A., Horn, T. & Uhlenhaut, N. H. Protocol for using heterologous spike-ins to normalize for technical variation in chromatin immunoprecipitation. STAR Protocols 2, 100609, doi:10.1016/j.xpro.2021.100609 (2021).

2 Bonhoure, N. et al. Quantifying ChIP-seq data: a spiking method providing an internal reference for sample-to-sample normalization. Genome Research 24, 1157-1168, doi:10.1101/gr.168260.113 (2014).

3 Chen, K. et al. The Overlooked Fact: Fundamental Need for Spike-In Control for Virtually All Genome-Wide Analyses. Molecular and Cellular Biology 36, 662-667, doi:10.1128/MCB.00970-14 (2015).